# OpenHOI: Open-World Hand-Object Interaction Synthesis with Multimodal Large Language Model

**Zhenhao Zhang[1], Ye Shi[1]\*, Lingxiao Yang[1], Suting Ni[1], Qi Ye[2], Jingya Wang [1]\***

[1]ShanghaiTech University   [2]Zhejiang University

{zhangzhh2024, shiye, yanglx2023, nist2024, wangjingya}@shanghaitech.edu.cn
qi.ye@zju.edu.cn

## Abstract

Understanding and synthesizing realistic 3D hand-object interactions (HOI) is critical for applications ranging from immersive AR/VR to dexterous robotics. Existing methods struggle with generalization, performing well on closed-set objects and predefined tasks but failing to handle unseen objects or open-vocabulary instructions. We introduce OpenHOI, the first framework for open-world HOI synthesis, capable of generating long-horizon manipulation sequences for novel objects guided by free-form language commands. Our approach integrates a 3D Multimodal Large Language Model (MLLM) fine-tuned for joint affordance grounding and semantic task decomposition, enabling precise localization of interaction regions (e.g., handles, buttons) and breakdown of complex instructions (e.g., "Find a water bottle and take a sip") into executable sub-tasks. To synthesize physically plausible interactions, we propose an affordance-driven diffusion model paired with a training-free physics refinement stage that minimizes penetration and optimizes affordance alignment. Evaluations across diverse scenarios demonstrate OpenHOI's superiority over state-of-the-art methods in generalizing to novel object categories, multi-stage tasks, and complex language instructions. Our project page at https://openhoi.github.io

## 1 Introduction

Hand-object interaction (HOI) involves jointly modeling hand articulation and object dynamics to generate and interpret realistic manipulation sequences [9, 17, 25, 26, 15, 47, 5]. This reflects one of the most pervasive human behaviors, deeply embedded in daily activities. Generating and understanding 3D HOI sequences is critical for advancing machine capabilities in human-centric applications. In augmented/virtual reality (AR/VR), realistic HOI modeling enables immersive digital experiences, allowing users to manipulate virtual objects naturally. For robotics, it provides the foundation for dexterous, feedback-driven manipulation in unstructured environments.

Generating HOI sequences from natural language instructions remains a significant challenge in 3D interaction research. While traditional methods [19, 40] rely on handcrafted motion priors, recent diffusion-based approaches [2, 3] directly map text to action sequences, eliminating the need for manual motion design. However, due to limited data and modeling capacity, these models only deal with closed datasets and struggle to generalize to unseen objects and open-vocabulary instructions.

Recent advances in Large Language Models (LLMs) have significantly enhanced joint vision-language understanding. Building on this progress, concurrent work like HOIGPT [13] proposes an LLM-based architecture for aligning textual instructions with HOI sequences. However, these approaches face

---

*Corresponding author.

39th Conference on Neural Information Processing Systems (NeurIPS 2025).

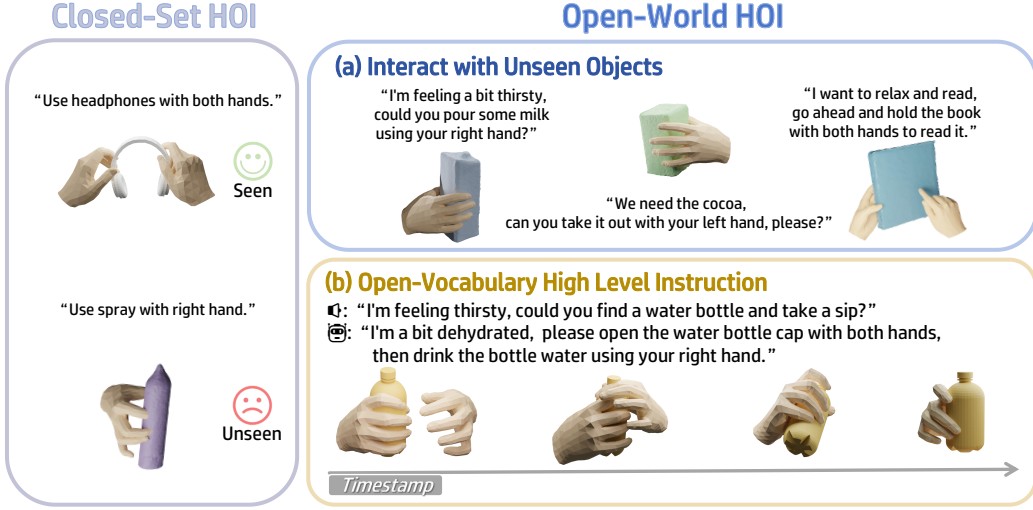

Figure 1: Motivation: **OpenHOI** introduces an open-world framework for generating HOI sequences that demonstrates strong generalization across seen and unseen objects, high-level instructions, and long-horizon tasks.

fundamental generalization challenges in unseen shapes and complex 3D interaction scenarios due to the lack of 3D knowledge in LLM. Fig. 1 illustrates the comparison between conventional Closed-Set HOI and Open-World HOI.

The growth of 3D datasets has advanced 3D multimodal large language models (MLLMs) with strong capabilities in understanding and grounding 3D geometric, semantic, and functional relationships. Critically, fine-grained part-level grounding and affordance grounding [34] [4] can identify actionable object regions and their interaction possibilities, providing strong priors for synthesizing physically consistent hand-object interactions (HOI). Building on this insight, we propose OpenHOI, the first open-world HOI synthesis framework capable of generating long-horizon manipulation sequences for unseen objects guided by open-vocabulary instructions. Our approach fine-tunes a 3D MLLM endowed with comprehensive affordance reasoning priors, enabling two core capabilities: 1) Generalizable Affordance Prediction: Precise localization of interaction regions (e.g., handles, buttons) for both known and novel objects. 2) Instruction Decomposition: Breaking down open-vocabulary commands (e.g., "I'm feeling thirsty, could you find a water bottle and take a sip") into executable sub-tasks grounded in object affordances. By integrating these priors with a diffusion-based interaction generation and physics-aware refinement, OpenHOI achieves unprecedented generalization across object categories, task horizons, and linguistic complexity.

In summary, our contributions are as follows:

- We introduce OpenHOI, the first open-world hand-object interaction synthesis framework capable of generating long-horizon manipulation sequences for unseen objects guided by open-vocabulary instructions. Unlike prior closed-set HOI methods, OpenHOI generalizes to novel objects and complex, linguistically diverse commands.

- OpenHOI involves fine-tuning a 3D MLLM that jointly learns geometric affordance priors and semantic task decomposition. Subsequently, affordance-driven HOI Diffusion with Physical Refinement is developed to generate realistic HOI sequences for each task.

- Our experiments demonstrate state-of-the-art performance, outperforming existing methods with a large margin. Notably, OpenHOI generalizes robustly to unseen objects and open-vocabulary instructions, achieving strong compositional generalization across diverse scenarios.

Table 1: Comparative analysis of our method versus existing HOI synthesis approaches.

| Methods | Open-world | 3D MLLM | Open-vocabulary | Long-horizon | Planning | Motion In-between |
|---|---|---|---|---|---|---|
| HandDiffuse [23] | | | | | | ✓ |
| Text2hoi [2] | | | | | | |
| HOIGPT [13] | | | ✓ | ✓ | | |
| Ours | ✓ | ✓ | ✓ | ✓ | ✓ | ✓ |

## 2 Related Work

**Hand-Object Interaction Synthesis.** Synthesizing realistic and diverse hand-object interactions (HOI) have gained significant attention, with numerous approaches exploring this problem under various settings. These include the creation of large-scale datasets like GigaHands [7] and OAKINK2 [53], methods for affordance learning [21, 56, 32, 45]. Others have focused on synthesizing two-hand interactions like InterHandGen [19] and HandDiffuse [23], or on physics-aware synthesis and dexterous grasping [44, 46, 57]. Several key works have specifically tackled text-guided or semantically rich HOI synthesis. Text2HOI [2] introduced a pioneering approach to generate 3D Hand-Object Interaction Sequences from text by decomposing the task into contact prediction and motion generation using a diffusion model, but it can struggle with fine-grained control from low-level text and tends to produce short HOI interaction sequences. HOIGPT [13] adopts an LLM-based sequential model to predict hand-object trajectories from text for long-horizon sequences, yet it lacks explicit affordance-guided mechanisms and cannot ensure smooth motion in-between multiple sub-sequences in complex interactions. SemGrasp [20] focuses on semantic-aware grasp generation, but it discretizes motion space and is limited to static grasps. Grasp as You Say [46] addresses category-level language-guided grasping, but it requires hand-crafted hand-object contact templates and lacks dynamics. While these methods have advanced the field, generating fine-grained, diverse, and long-horizon hand-object interaction sequences with high-level text on unseen objects cohesively remains a significant challenge.

**Multimodal Large Language Model.** The advent of Large Language Models (LLMs) has revolutionized natural language understanding, and their influence is gradually expanding into 3D perception and interaction tasks. Some recent works leverage pure LLM architectures to handle motion generation from language. For instance, HOIGPT [13] and MotionGPT [16] adopt VQ-VAE tokenization and GPT-style architectures to achieve bidirectional generation between natural language and HOI systhesis. However, these models operate primarily on discrete motion tokens, and lack grounding in actual 3D perception or reasoning, they cannot perceive objects or environments in 3D nor reason about spatial affordances or constraints, thus limiting their ability to generate context-aware interactions. Conversely, Multimodal Large Language Models (MLLMs) have emerged as powerful tools that extend the success of LLMs into various domains, aiming to bridge the gap between language and other modalities such as images, videos, and 3D representations, enabling tasks like visual question answering like ShapeLLM [34], 3D object generation like Point-E [29] and CLIP-Forge [37], and affordance understanding like LASO [22], DAG[42]and AffordanceLLM [35]. These MLLMs demonstrate strong capabilities in static 3D understanding, especially in segmenting or identifying affordances from language prompts. Other MLLM-based approaches like SeqAfford [52] and 3D-AffordanceLLM [4] further extend this to sequential or compositional affordance reasoning. However, these approaches remain limited to perception tasks and do not address the synthesis of continuous hand-object interactions or motion dynamics. In contrast, while LLM-based methods focus on generating motion from text without grounding in 3D environments, and MLLM-based methods enable 3D spatial understanding without supporting interaction generation, none of the existing approaches bridge both capabilities. Our work addresses this gap by leveraging an MLLM that is trained to jointly model language, 3D perception, and interaction dynamics. This enables our model to both understand object affordances from language and synthesize long-horizon, task-consistent hand-object interactions with unseen objects from open-vocabulary instructions. By unifying reasoning and generation, our approach extends 3D MLLMs toward fine-grained, dynamic, and open-world HOI synthesis.

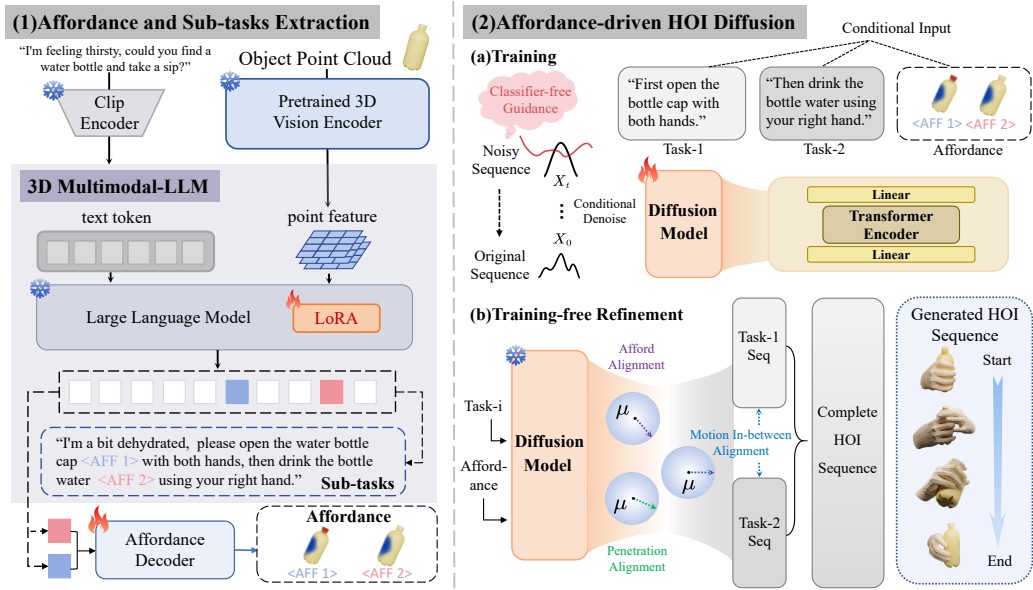

Figure 2: Pipeline: Our framework comprises two sequential components. First, a 3D multimodal large language model (3D MLLM) ingests high-level instructions and object point clouds to generate sequential affordance maps and decompose the high-level task into a sequence of sub-tasks. Second, the diffusion model takes the affordance map and the decomposed task sequence as conditions to synthesize realistic hand-object interaction sequences.

## 3 Method

We propose OpenHOI, a novel framework for generating long-horizon hand-object interaction sequences guided by high-level open-vocabulary instructions. OpenHOI generalizes to unseen object categories by leveraging semantic reasoning from a 3D MLLM paired with 3D affordance prior. Our framework consists of two core components:

### 3.1 Instruction Decomposition and Affordance Reasoning via 3D MLLM

Given a 3D object point cloud and a high-level, free-form instruction (e.g., "I am thirsty"), the 3D MLLM semantically grounds the instruction into object-centric affordances by bridging the gap between abstract intent and actionable object functionality. This process yields two structured outputs: (1) A spatial affordance map that identifies geometrically plausible interaction regions (e.g., highlighting the cap of a water bottle for opening and the body for grasping). (2) A temporally decomposed sub-task sequence that translates the instruction into executable atomic actions (e.g., 1) grasp bottle cap with both hands → 2) twist counterclockwise to open → 3) lift bottle to mouth with right hand").

**Network structure of 3D MLLM.**  Recent advances in 3D multi-modal language modeling have significantly enhanced open-world understanding of 3D objects. In particular, ShapeLLM [34] has been pretrained to capture a wide range of embodied interactions. Consequently, we adopt ShapeLLM as our backbone. Its point-cloud encoder ReCon++ is pretrained via multi-view distillation from ReCon [33], and its language component is initialized from LLaMa [41]. Prior approaches to 3D affordance prediction have typically relied on standalone 3D backbones [51] or on separate point-to-language encoders [22], which often lack robust reasoning and open-world generalization capabilities. By leveraging a unified 3D MLLM rather than exclusively using pure LLMs or conventional visual architectures, we achieve both enhanced generalization to unseen objects and affordances and the intrinsic integration of affordance perception into natural language representations, thereby facilitating subsequent affordance reasoning.

**In-context 3D Affordance Reasoning.** Although 3D MLLMs effectively align 3D representations with natural language, they are predominantly tailored for object-centric text generation and thus lack inherent support for dense 3D prediction tasks such as fine-grained affordance segmentation. To address this limitation, we augment the MLLM vocabulary with a dedicated segmentation token <AFF>, following the design of Lisa [18], thereby enabling the model to represent and reason about segmentation outputs within its language-based framework. Formally, given a point-cloud input $\mathbf{F}_{\text{obj}}$ and instruction text $\mathbf{T}_{\text{ins}}$ expressing user intent over candidate objects, the 3D MLLM jointly encodes both modalities to produce a sub-task sequence:

$$\tilde{\mathbf{T}}_{\text{sub\_tasks}} = \text{MLLM}\big(\mathbf{F}_{\text{obj}}, \mathbf{T}_{\text{ins}}\big), \tag{1}$$

which contains $S$ occurrences of the segmentation token <AFF>, each marking a predicted affordance region. We then collect the corresponding last-layer embeddings $\{\mathbf{h}_{\text{aff}}^{(i)}\}_{i=0}^{S-1}$ and pass each through an affordance decoder:

$$\tilde{\mathbf{A}}_{\text{obj}}^{(i)} = \text{Decoder}_{\text{aff}}\big(\mathbf{F}_{\text{obj}}, \mathbf{h}_{\text{aff}}^{(i)}\big), \quad i = 0, \ldots, S-1. \tag{2}$$

**Coarse-to-Fine Affordance Tuning.** Our training pipeline comprises two sequential stages. First, we fine-tune the model on large-scale, coarse-grained, object-centric static affordance datasets [35, 52], to instill strong affordance priors, thereby enhancing its generalization and open-vocabulary capabilities. Second, we fine-tune the stage 1 model on a smaller dynamic hand-object interaction dataset, which can get fine-grained affordance maps by using the voxel-based method, to generate more precise affordance Maps and task decomposition sequences. We leverage an auto-regressive cross-entropy loss $L_{\text{task}}$ to supervise sub-tasks generation, complemented by Dice loss and Binary Cross-Entropy loss $L_{\text{aff}}$ to guide affordance prediction.

$$L = \lambda_{\text{task}} L_{\text{task}}(\mathbf{Y}_{\text{sub\_tasks}}, \tilde{\mathbf{Y}}_{\text{sub\_tasks}}) + \lambda_{\text{aff}} L_{\text{aff}}(\mathbf{A}_{\text{obj}}, \tilde{\mathbf{A}}_{\text{obj}}), \tag{3}$$

where the weights $\lambda_{\text{task}}, \lambda_{\text{aff}}$ are utilized to balance the different loss items.

### 3.2 Affordance-driven HOI Diffusion with Physical Refinement

To generate long-horizon, natural, and physically plausible hand-object interaction sequences $X$ based on **Interaction Prior C** from Sec 3.1, which include affordance prior, sub-tasks, and object pointcloud as follows,

$$\mathbf{C} = [\tilde{\mathbf{A}}_{\text{obj}}, f^{\text{clip}}(\tilde{\mathbf{T}}_{\text{sub\_tasks}}), \mathbf{F}_{\text{obj}}], \tag{4}$$

We propose **Affordance-driven HOI Diffusion**, which decomposes this challenging task into tractable components. During training, we improve the alignment of the affordance map via classifier-free guidance [11, 30, 12]. During inference, we propose a training-free refinement method that generates natural, physically plausible, and long-horizon HOI sequences.

**Training HOI Diffusion with Affordance Prior from MLLM.** During training, given a conditioning pair $[X, \mathbf{C}]$, we first add noise to $X$ using the forward diffusion process:

$$p(X_t | X_0) \sim \mathcal{N}(\sqrt{\bar{\alpha}_t} X_0, (1 - \bar{\alpha}_t)I), \tag{5}$$

where $X_0$ denotes the original HOI sequence, $X_t$ is its noisy version at diffusion timestep $t$, and $\alpha_t$ controls the noise schedule. We then train a transformer-based neural network $\hat{X}_\theta$ to directly predict the HOI sequence $X_0$ from the noisy input. The training objective for HOI generation is defined as:

$$L_{\text{hoi\_train}}(\hat{X}_\theta, \mathbf{C}) = L_{\text{hoi\_diff}}(\hat{X}_\theta, \mathbf{C}) + L_{\text{hoi\_distance}}(\hat{X}_\theta) + L_{\text{hoi\_orient}}(\hat{X}_\theta), \tag{6}$$

where $L_{\text{hoi\_diff}}(\hat{X}_\theta, \mathbf{C}) = \mathbb{E}_{X_0 \sim p(X_0), X_t \sim p(X_t | X_0), t \sim [1,T]} ||X_0 - X_\theta(X_t, t, \mathbf{C})||_2^2$ is the denoising loss of the diffusion process for HOI reconstruction. $L_{\text{hoi\_distance}}(\hat{X}_\theta)$ donates the HOI distance map loss, which encourages precise surface contact and enhances the physical plausibility of the contact area by penalizing errors more strongly when the hand is near the object. $L_{\text{hoi\_orient}}(\hat{X}_\theta)$ represents the HOI relative orientation loss, which aligns predicted orientations with ground truth to ensure accurate rotational poses for tasks like grasping and manipulation by modeling hand–object relative rotation, respectively. To better align the generated HOI sequences with the Interaction Prior $\mathbf{C}$, we adopt the classifier-free guidance strategy:

$$X_\theta^s(X_t, t, \mathbf{C}) = X_\theta(X_t, t, \emptyset) + s \cdot (X_\theta(X_t, t, \mathbf{C}) - X_\theta(X_t, t, \emptyset)), \tag{7}$$

where $s$ is the guidance scale. To enable this strategy, we randomly mask 10% of the conditional inputs $\mathbf{C}$ during training to train the unconditional [8] model $X_\theta(X_t, t, \emptyset)$ alongside the conditional one. During sampling, we gradually denoise $X_t$ to $X_0$ via posterior distribution $p(X_{t-1}|X_\theta, X_t)$:

$$X_{t-1} = \mu_t + \sigma_t \epsilon, \epsilon \sim \mathcal{N}(0, I), \tag{8}$$

where $\mu_t = \frac{\sqrt{\alpha_t}(1-\bar{\alpha}_{t-1})}{1-\bar{\alpha}_t}X_t + \frac{\sqrt{\bar{\alpha}_{t-1}}\beta_t}{1-\bar{\alpha}_t}X_\theta^s(X_t, t, \mathbf{C})$, and $\sigma_t = \frac{1-\bar{\alpha}_{t-1}}{1-\bar{\alpha}_t}\beta_t$.

**Training-free HOI Diffusion with Physical Refinement.** While the vanilla diffusion sampling process in Equation 8 models the joint conditional distribution of HOI sequences, it suffers from two key limitations: 1) **Lack of Physical Constraints.** The sampling process does not inherently enforce physical plausibility [28], such as preventing interpenetration or ensuring stable contact. While prior work addresses this by training discriminative physical refiners, such methods often deviate from the underlying joint HOI distribution, introducing artifacts due to distribution shift. 2) **Temporal Incoherence.** The generated sequences exhibit discontinuous hand motions, particularly during transitions between sub-sequences, leading to unrealistic motion dynamics.

To address these limitations, we propose three key refinement objectives during the diffusion sampling process: 1) affordance refinement for precise contact, 2) physical constraint refinement to prevent penetration, and 3) temporal coherence refinement for smooth transitions, all achieved without additional training or distribution-shifting.

a) *Affordance Refinement:* Since direct affordance-based HOI generation often fails to produce precise hand-object contact, we propose an affordance-aware loss to guarantee geometric consistency:

$$l_{\text{aff}} = \mathbb{1}_{\text{left}} \cdot \|d(\hat{J}_{\text{lhand}}, \tilde{P}_{\text{obj}}^{\text{ljoint}})\|^2 + \mathbb{1}_{\text{right}} \cdot \|d(\hat{J}_{\text{rhand}}, \tilde{P}_{\text{obj}}^{\text{rjoint}})\|^2, \tag{9}$$

where $d(\cdot, \cdot)$ is the Euclidean distance between hand joints $(\hat{J}_{\text{lhand}}, \hat{J}_{\text{rhand}})$ and closest affordance region $(\tilde{P}_{\text{obj}}^{\text{ljoint}}, \tilde{P}_{\text{obj}}^{\text{rjoint}})$.

b) *Penetration Refinement:* To mitigate interpenetration artifacts, we propose a penetration loss:

$$l_{\text{penetration}} = \mathbb{1}_{\text{left}} \cdot \|d(\hat{V}_{\text{lhand}}, \tilde{P}_{\text{obj}}^{\text{lvert}})\|^2 + \mathbb{1}_{\text{right}} \cdot \|d(\hat{V}_{\text{rhand}}, \tilde{P}_{\text{obj}}^{\text{rvert}})\|^2, \tag{10}$$

where $d(\cdot, \cdot)$ is the Euclidean distance between hand joints $(\hat{V}_{\text{lhand}}, \hat{V}_{\text{rhand}})$ and closest affordance region $(\tilde{P}_{\text{obj}}^{\text{lvert}}, \tilde{P}_{\text{obj}}^{\text{rvert}})$.

c) *Motion In-between Refinement:*

For seamless transitions between HOI sub-sequences, we synthesize natural hand motion $\hat{V}_{\text{trans}}^{0:T}$ bridging the end pose $V_{\text{pre}}^T$ of the preceding sequence and the start pose $V_{\text{after}}^0$ of the subsequent sequence. The transition loss can be defined as:

$$l_{\text{transition}} = ||\hat{V}_{\text{trans}}^0 - V_{\text{pre}}^T||_2^2 + ||\hat{V}_{\text{trans}}^T - V_{\text{after}}^0||_2^2, \tag{11}$$

where $\hat{V}_{\text{trans}}^0, \hat{V}_{\text{trans}}^T$ denote the predicted start and end frames of the transition motion.

While gradient descent after each denoising step could minimize these losses, it risks introducing artifacts and distribution shifts [49]. Recently, a training-free conditional diffusion model named DSG [50] offers larger, adaptive step sizes to the loss function to achieve better alignment with the constraints while preserving the original distribution learned by the diffusion model. Inspired by DSG, we also introduce the *Spherical Gaussian Constraint* during the sampling stage to preserve the original distribution, thus mitigating the distribution-shifting problem. We utilize the analytical solution to enforce steepest gradient descent to enhance alignment:

$$D^\star = -\sqrt{d}\sigma_t \nabla_{X_t} l(X_\theta^s(X_t, t, c)), \tag{12}$$

where $d$ represents the data dimensions and $l$ denotes the loss in Eq 9, 10, 11 for refinement stage. To enhance the sample quality, we utilize a mixture of deterministic steepest gradient descent direction and random sampling direction:

$$D_{\text{mix}} = D_{\text{sample}} + w \cdot (D^\star - D_{\text{sample}}), \tag{13}$$

$$X_{t-1} = \mu_t + \sqrt{d}\sigma_t \frac{D_{\text{mix}}}{||D_{\text{mix}}||} \tag{14}$$

where $D_{\text{sample}} = \sigma_t \epsilon_t$ is the random sampling direction, and $w$ represents the guidance rate.

# 4 Experiments

Our framework integrates two core stages: 1) fine-tuning a 3D MLLM to predict object affordance maps and decomposing open-vocabulary instructions into concrete sub-tasks, and 2) synthesizing HOI sequences with the condition output of stage 1. We evaluate our method using diverse datasets and metrics, demonstrating its capability to generate long-horizon HOI sequences with high-level instructions of both seen and unseen objects. Comparative experiments and ablations validate our design choices. Our experiments were conducted on NVIDIA A100 GPU.

## 4.1 Dataset

For our experiments, we utilize two prominent hand-object interaction datasets: **GRAB** [38], which provides comprehensive full-body motion data of subjects interacting with 51 everyday objects, and **ARCTIC** [6], a large-scale dataset specializing in bi-manual interactions with articulated objects and dense 3D annotations. We preprocess both two datasets with a unified pipeline, which involves initial geometric operations on the object point clouds—specifically, upsampling to support detailed affordance map generation via our inference model, followed by downsampling to ensure computational efficiency during training. Crucially, we enrich the original annotations by employing a multimodal large language model (MLLM) to convert low-level HOI motion descriptions into open-vocabulary, intent-centric language instructions. These semantic labels enhance the expressiveness of the data and allow our model to better generalize to unseen scenarios, supporting open-world HOI synthesis with long-horizon sequences and diverse objects. For both GRAB [38] and ARCTIC [6], we follow a standard protocol by partitioning each dataset into 80% for training and 20% for unseen testing, ensuring reliable evaluation of our model's generalization capabilities. This setup allows us to rigorously assess performance on unseen objects, motions, and interaction intents, validating the robustness of our MLLM-guided generation framework across both single-hand and bi-manual interaction scenarios.

## 4.2 Evaluation metric

To comprehensively evaluate the quality, diversity, realism, and physical plausibility of our generated hand-object interaction (HOI) sequences, we employ a multi-faceted set of quantitative and qualitative metrics inspired by prior works [2, 13, 24, 27, 40, 14]. We roughly divide the evaluation indicators into three categories

- **Motion Accuracy.** We evaluate geometric precision using the Mean Per-Joint Position Error (**MPJPE**;) computed over hand joints, and assess object placement with the Final Object Location Error (**FOL**;), defined as the Euclidean distance between the predicted object center and the target location at the final frame.
- **Generation Realism.** We measure realism via the Fréchet Inception Distance (**FID**;) between real and synthesized motions in a pre-trained motion feature space, capturing distributional alignment and perceptual fidelity.
- **Diversity & Multi-modality. Diversity** quantifies across-prompt variability of generated outputs, while **MModality** captures within-prompt variability across multiple samples. Both are computed from pairwise distances or variance statistics in the motion feature space.

Lower MPJPE, FOL, and FID indicate higher accuracy and fidelity, while higher multi-modality and closer to GT diversity reflect stronger generative expressiveness.

## 4.3 Main Results

**Comparison with SOTA Methods.** We evaluate our method under the widely adopted seen / unseen split protocol and compare it against state-of-the-art methods (i.e., MDM [39], TM2T [10], MotionGPT [16], Text2HOI [2] on both GRAB [38] and ARCTIC [6] datasets. As shown in Table 2 and Table 3, our method consistently outperforms all baselines across both seen and unseen objects.

These results firmly establish our method as state-of-the-art, demonstrating the strong generalization ability of our MLLM-guided Affordance Reasoning and Affordance-driven HOI diffusion with Physical Refinement generation, which validates the effectiveness of our Open-World HOI synthesis

framework in generating long-horizon HOI sequences of unseen objects from Open-vocabulary instructions.

Table 2: **Main Results on GRAB.**

| | Method | MPJPE↓ | FOL↓ | FID ↓ | Diversity → | MModality ↑ |
|---|---|---|---|---|---|---|
| | GT | - | - | - | 4.66 | - |
| **Seen** | MDM[39] | 74.92±2.25 | 0.62±0.02 | 62.37±1.56 | 3.28±0.10 | 12.77±0.45 |
| | TM2T[10] | 59.27±1.19 | 0.46±0.06 | 57.41±2.30 | 3.60±0.07 | 21.28±0.82 |
| | MotionGPT[16] | 63.94±2.56 | 0.43±0.01 | 52.03±1.82 | 3.61±0.08 | 20.26±0.51 |
| | Text2HOI[2] | 56.29±2.13 | 0.44±0.03 | 33.72±1.27 | 3.41±0.16 | 17.71±0.87 |
| | Ours | **47.64±1.03** | **0.26±0.02** | **26.43±0.77** | **3.69±0.27** | **24.59±2.01** |
| **Unseen** | MDM[39] | 92.97±1.86 | 0.69±0.03 | 75.59±1.89 | 3.07±0.11 | 11.15±0.85 |
| | TM2T[10] | 61.07±1.34 | 0.55±0.02 | 66.43±1.66 | 3.37±0.07 | 14.03±0.67 |
| | MotionGPT[16] | 66.26±1.99 | 0.51±0.01 | 56.49±1.98 | 2.852±0.07 | 16.36±0.53 |
| | Text2HOI[2] | 60.67±1.80 | 0.41±0.02 | 36.96±0.77 | 1.80±0.05 | 10.98±0.44 |
| | Ours | **51.34±0.85** | **0.27±0.01** | **28.29±0.62** | **3.61±0.09** | **19.91±0.63** |

Table 3: **Main Results on ARCTIC.**

| | Method | MPJPE↓ | FOL↓ | FID ↓ | Diversity → | MModality ↑ |
|---|---|---|---|---|---|---|
| | GT | - | - | - | 3.39 | - |
| **Seen** | MDM[39] | 72.67±0.63 | 0.60±0.05 | 33.66±0.19 | 2.35±0.05 | 8.20±0.20 |
| | TM2T[10] | 54.39±0.64 | 0.41±0.04 | 34.12±0.49 | 1.67±0.02 | 13.60±0.17 |
| | MotionGPT[16] | 60.17±0.72 | 0.41±0.03 | 31.58±0.46 | 1.89±0.02 | 13.23±0.09 |
| | Text2HOI[2] | 52.16±0.41 | 0.33±0.01 | 23.35±0.33 | 2.43±0.02 | 11.21±0.20 |
| | Ours | **45.15±0.94** | **0.25±0.04** | **19.74±0.16** | **2.65±0.03** | **15.25±1.44** |
| **Unseen** | MDM[39] | 86.75±1.35 | 0.64±0.01 | 41.53±1.37 | 1.58±0.04 | 7.13±0.63 |
| | TM2T[10] | 55.57±1.26 | 0.53±0.03 | 37.22±0.75 | 1.54±0.12 | 11.23±0.44 |
| | MotionGPT[16] | 64.41±0.73 | 0.43±0.04 | 33.99±2.43 | 1.50±0.09 | 11.08±0.79 |
| | Text2HOI[2] | 57.83±1.61 | 0.39±0.01 | 25.22±0.59 | 1.61±0.06 | 7.11±0.25 |
| | Ours | **47.25±0.39** | **0.28±0.03** | **20.05±0.80** | **2.49±0.08** | **12.66±0.71** |

**Qualitative results.** We present the generated HOI sequence results. Open-world capability enables the generation of long-horizon HOI sequences under both unseen-object and open-vocabulary conditions. In this section, long-horizon HOI results are shown in Fig. 3. Experimental findings demonstrate physical realism and coherence of the generated sequences.

## 4.4 Ablation Study

We conduct systematic ablations on the GRAB [38] and ARCTIC [6] dataset to validate the necessity of our core components through controlled experiments. Each component is carefully analyzed to demonstrate its contribution to robust hand-object interaction generation, the details can be found in Table 4 and Table 5.

**Affordance Awareness (w/o Affordance).** We remove the part of obtaining accurate affordance maps by a well-trained MLLM, which significantly degrades interaction quality on each evaluation metric. Without this component, the model loses its ability to focus on functionally critical object regions, leading to unnatural hand placements and increased penetrations. This confirms that explicit affordance grounding is essential for semantically meaningful interactions.

**Classifier-Free Guidance Diffusion (w/o CFG).** Classifier-Free Guidance(CFG) plays a critical role in ensuring that the generated hand-object interaction (HOI) sequences remain aligned with the input conditions $\mathbf{C}$. Without CFG, the model often fails to adhere to the conditioning signal, leading to semantic misalignment and affordance mismatch.

**Loss-guided Physical Refinement (w/o $l_{\text{penetration}}$, w/o $l_{\text{aff}}$).** Experimental result demonstrates the critical importance of our loss-guidance strategy for temporal coherence and physical plausibility.

Figure 3: Qualitative result: The visualization results showcase three types of long-horizon sequences—seen-object, unseen-object, and multi-object. The experiments demonstrate that our method exhibits strong generalization on both unseen objects and open-vocabulary instructions, enabling open-world HOI sequence synthesis.

Without this guidance, the diffusion model exhibits two key failure modes: first, it fails to generate complete long-horizon HOI sequences, particularly in complex bimanual interactions where discrete actions remain disjointed rather than forming fluid motions; second, it produces physically unrealistic results with frequent object penetrations ($l_{\text{penetration}}$) and unnatural affordance patterns ($l_{\text{aff}}$), most noticeable during precision manipulation tasks. Through direct optimization of these constraints via our loss-guidance framework, we achieve significant improvements in generating physically valid and temporally coherent hand-object interactions.

Table 4: **Ablation Study on GRAB**

| | Method | MPJPE↓ | FOL↓ | FID↓ | Diversity→ | MModality↑ |
|---|---|---|---|---|---|---|
| | GT | - | - | - | 4.66 | – |
| **Seen** | w/o Affordance | 56.04 ± 1.22 | 0.36 ± 0.03 | 31.60 ± 0.93 | 3.47 ± 0.05 | 16.89 ± 1.50 |
| | w/o CFG | 51.54 ± 0.80 | 0.41 ± 0.02 | 29.52 ± 0.90 | 3.39 ± 0.18 | 20.55 ± 1.80 |
| | w/o $l_{penetration}$ | 53.42 ± 0.37 | 0.42 ± 0.01 | 28.43 ± 0.74 | 3.31 ± 0.21 | 18.59 ± 1.74 |
| | w/o $l_{aff}$ | 49.17 ± 0.36 | 0.43 ± 0.02 | 29.20 ± 0.85 | 3.41 ± 0.21 | 19.25 ± 1.90 |
| | Ours | **47.64±1.03** | **0.26±0.02** | **26.43±0.77** | **3.69±0.27** | **24.59±2.01** |
| **Unseen** | w/o Affordance | 60.37 ± 1.69 | 0.40 ± 0.08 | 36.82 ± 1.04 | 3.34 ± 0.05 | 15.72 ± 1.50 |
| | w/o CFG | 55.04 ± 0.83 | 0.41 ± 0.06 | 37.05 ± 1.38 | 3.24 ± 0.03 | 18.46 ± 1.13 |
| | w/o $l_{penetration}$ | 56.68 ± 1.6 | 0.40 ± 0.11 | 35.69 ± 1.65 | 3.36 ± 0.19 | 18.78 ± 0.38 |
| | w/o $l_{aff}$ | 54.56 ± 0.47 | 0.37 ± 0.19 | 36.13 ± 0.72 | 3.28 ± 0.37 | 18.29 ± 0.38 |
| | Ours | **51.34±0.85** | **0.27±0.01** | **28.29±0.62** | **3.61±0.09** | **19.91±0.63** |

Table 5: **Ablation Study on ARCTIC**

| | Method | MPJPE↓ | FOL↓ | FID↓ | Diversity→ | MModality↑ |
|---|---|---|---|---|---|---|
| | GT | - | - | - | 3.39 | – |
| **Seen** | w/o Affordance | $53.03 \pm 2.76$ | $0.40 \pm 0.03$ | $26.21 \pm 0.54$ | $2.03 \pm 0.19$ | $13.68 \pm 0.37$ |
| | w/o CFG | $52.79 \pm 0.95$ | $0.38 \pm 0.02$ | $25.44 \pm 0.29$ | $3.39 \pm 0.18$ | $14.05 \pm 2.72$ |
| | w/o $l_{penetration}$ | $51.66 \pm 0.88$ | $0.37 \pm 0.01$ | $26.73 \pm 0.37$ | $2.45 \pm 0.28$ | $13.99 \pm 0.58$ |
| | w/o $l_{aff}$ | $46.35 \pm 1.13$ | $0.39 \pm 0.02$ | $25.08 \pm 0.20$ | $2.31 \pm 0.07$ | $13.06 \pm 1.34$ |
| | Ours | $\mathbf{45.15 \pm 0.94}$ | $\mathbf{0.25 \pm 0.04}$ | $\mathbf{19.74 \pm 0.35}$ | $\mathbf{2.65 \pm 0.16}$ | $\mathbf{15.25 \pm 1.44}$ |
| **Unseen** | w/o Affordance | $57.29 \pm 2.33$ | $0.43 \pm 0.03$ | $30.05 \pm 0.71$ | $1.97 \pm 0.36$ | $11.22 \pm 0.60$ |
| | w/o CFG | $56.25 \pm 0.95$ | $0.45 \pm 0.02$ | $29.44 \pm 0.62$ | $2.25 \pm 0.24$ | $11.45 \pm 1.03$ |
| | w/o $l_{penetration}$ | $55.66 \pm 0.14$ | $0.41 \pm 0.03$ | $27.51 \pm 1.04$ | $2.13 \pm 0.10$ | $10.58 \pm 0.39$ |
| | w/o $l_{aff}$ | $49.18 \pm 1.13$ | $0.42 \pm 0.02$ | $27.36 \pm 0.25$ | $2.08 \pm 0.13$ | $11.01 \pm 1.02$ |
| | Ours | $\mathbf{47.25 \pm 0.39}$ | $\mathbf{0.28 \pm 0.03}$ | $\mathbf{20.05 \pm 0.80}$ | $\mathbf{2.49 \pm 0.08}$ | $\mathbf{12.66 \pm 0.71}$ |

## 4.5 Failure cases and error analysis

Given a scenario, there is a row of cabinets, and the instruction "Open the cabinet", our model can open one but cannot target a specific one (e.g., "Open the second cabinet"). This limitation arises because the model has not been trained on a large-scale 3D QA dataset, resulting in reduced logical reasoning capabilities. Due to the accumulation of model errors, performance typically degrades after more than three consecutive actions(more than 450 frames).

## 5 Conclusion

We present OpenHOI, the first open-world framework for synthesizing long-horizon 3D hand-object interaction (HOI) sequences guided by open-vocabulary instructions. By fine-tuning a 3D multimodal large language model (MLLM) to jointly model geometric affordances and decompose semantic instructions, our method achieves strong generalization to unseen objects and linguistically complex tasks. The integration of affordance-driven diffusion-based generation and physics-aware refinement enables physically consistent manipulation sequences, advancing beyond closed-set HOI synthesis methods. Extensive evaluations demonstrate OpenHOI's superiority in handling novel object categories, multi-stage tasks, and open-ended language commands, bridging critical gaps in human-centric AI applications. While OpenHOI represents a significant step toward open-world HOI synthesis, several challenges remain: Although our physics-aware refinement improves interaction plausibility, fine-grained dynamics (e.g., fluid simulation for pouring tasks) remain challenging. Hybrid neuro-symbolic physics models may enhance realism. While OpenHOI supports multi-stage tasks, handling compositional long-horizon sequences like "cook a meal" remains challenging due to limitations in hierarchical task decomposition. Future work could explore chain-of-thought reasoning to better model such complex action hierarchies.

## Acknowledgement

This work was supported by National Natural Science Foundation of China (62406195, 62303319), Shanghai Local College Capacity Building Program (23010503100), ShanghaiTech AI4S Initiative SHTAI4S202404, HPC Platform of ShanghaiTech University, and MoE Key Laboratory of Intelligent Perception and Human-Machine Collaboration (ShanghaiTech University), and Shanghai Engineering Research Center of Intelligent Vision and Imaging. This work was also supported in part by computational resources provided by Fcloud CO., LTD.

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

# OpenHOI: Open-World Hand-Object Interaction Synthesis with Multimodal Large Language Model
## Appendix

**Appendix:**

## A  Implementation Details

**3D MLLM.**  We initialize our model from the ShapeLLM-7B checkpoint, freezing its 3D encoder and augmenting the visual backbone with Uni3D for robust dense 3D prediction, while the projection head is implemented as a shallow MLP, and LoRA is applied to streamline fine-tuning. Training unfolds in two stages: first, we optimize for seven epochs with AdamW (learning rate $2 \cdot 10^{-4}$, zero weight decay) under a cosine-annealing schedule and a 2% linear warm-up; then, we continue for three additional epochs with AdamW (learning rate $5 \cdot 10^{-4}$, zero weight decay) under the same cosine schedule but a 1% warm-up.

**Diffusion Model.** We employ a T = 1000-step noising process with a cosine noise schedule, and inject positional information at both the frame- and agent-levels using sinusoidal encodings. During sampling, we apply classifier-free guidance by randomly substituting 10% of conditioning inputs with unconditional noise while retaining 90% of the original conditions, and use a guidance scale of 2.5 to steer the denoising trajectory.

# B  Instruction Decomposition and Affordance Reasoning via 3D MLLM

## B.1  Affordance Reasoning

**3D Object Point Cloud Encoding.** We take as input a point cloud of an object, sampled to $N$ points. The backbone is a ReCon++ [34] network (or a similar architecture) that processes these points and produces per-point feature representations

$$F_{\text{obj}} \in \mathbb{R}^{N \times C},$$

which capture both local geometric details and the overall global context.

**Multi-Token Fusion Mechanism.** Rather than generating a single `<AFF>` segmentation token, PixelLM[36] defines, at each visual scale $\ell$, a segmentation codebook comprising $N$ learnable `<AFF>` tokens. After encoding the textual prompt, the model sequentially outputs the $N$ tokens, each associated with a hidden vector $h_i^\ell$. A linear projection $\phi$ then aggregates these vectors into a unified representation

$$h^\ell = \phi\big(h_1^\ell, \ldots, h_N^\ell\big),$$

which is concatenated with the scale-specific image features and fed into the pixel decoder to produce the final segmentation mask. Experiments on the MUSE validation set indicate that increasing $N$ from 1 to 3 improves cIoU, demonstrating that the multi-token fusion mechanism captures more nuanced semantic details and significantly enhances fine-grained segmentation.

## B.2  Instruction Decomposition

OpenHOI decomposes a single high-level instruction into an ordered sequence of actionable affordance steps. Each step is marked by a special token and then grounded spatially in the 3D point cloud.

**Instruction Text Encoding.** We take a natural-language instruction $T_{\text{ins}}$ as the model input. The backbone is a LLaMA-style Transformer that produces token-wise hidden states $\{h_t\}_{t=1}^T$ and aggregates them via a pooling operation into a single embedding

$$h_{\text{cls}} \in \mathbb{R}^D,$$

which is then used for downstream tasks.

**Segmentation Token Injection.** Extend the MLLM's vocabulary by adding a special marker `<AFF>`, which explicitly denotes the boundary of each sub-task in the generated sequence.

**Conditioned Autoregressive Generation.** Given the fused 3D point features $F_{\text{obj}}$ and the instruction embedding $h_{\text{cls}}$, the Transformer predicts an interleaved stream of action words and `<AFF>` tokens, for example:

$$\text{Pick} \rightarrow \text{<AFF>} \quad \text{Twist} \rightarrow \text{<AFF>} \quad \text{Lift} \rightarrow \text{<AFF>}$$

Let $S$ be the total number of `<AFF>` tokens generated.

**Boundary Localization & Hidden-State Extraction.** Record the positions $t_1, \ldots, t_S$ where `<AFF>` appears. For each $i = 1, \ldots, S$, extract the corresponding last-layer hidden vector

$$z_i = h_{t_i} \in \mathbb{R}^D,$$

which encodes the full context immediately preceding the end of sub-task $i$.

**Sequential Mask Decoding.** For each step $i$, use the query $E_i$ to perform cross-attention over the point features $F_{\text{obj}}$ and decode a per-point mask:

$$M_i(p) = \sigma\big(\text{Decoder}(\widetilde{F}_p, z_i)\big) \quad \text{for } p = 1, \dots, N,$$

where $\widetilde{F}$ are the fused features and $\sigma$ is the sigmoid activation. Collect the ordered set $\{M_1, M_2, \dots, M_S\}$ to obtain the final sequence of affordance masks, each aligned with its corresponding sub-task.

## B.3   Diffusion Process

Our framework employs diffusion models to learn the conditional distribution $p(X|\mathbf{C})$, of hand-object interaction (HOI) sequences, where the conditioning signal $\mathbf{C}$ combines:

- Object affordance prior $\tilde{\mathbf{A}}_{\text{obj}}$
- Sub-task embedding $f^{\text{clip}}(\tilde{\mathbf{T}}_{\text{sub\_tasks}})$
- Object point cloud features $\mathbf{F}_{\text{obj}}$

**Forward Process.** The diffusion process gradually corrupts the input data through the forward process with a fixed noise schedule $\alpha_t \in [0, T]$

$$p(X_t|X_0) \sim \mathcal{N}(\sqrt{\bar{\alpha}_t}X_0, (1 - \bar{\alpha}_t)I). \tag{15}$$

where $X_0$ is the original HOI sequence, $X_t$ represents its noisy version at timestep $t$ and $\bar{\alpha}_t = \prod_{i=1}^{t} \alpha_t$. This forward process progressively transforms the data distribution into a tractable Gaussian distribution $\mathcal{N}(0, I)$.

**Loss Function.** Like VAEs, the diffusion model can be optimized by maximizing the ELBO:

$$\log p_\theta(X_0|\mathbf{C}) = \log \int p_\theta(X_{0:T}|\mathbf{C})dX_{1:T} \tag{16}$$

$$= \log \int \frac{p_\theta(X_{0:T}|\mathbf{C})p(X_{1:T}|X_0, \mathbf{C})}{p(X_{1:T}|X_0, \mathbf{C})}dX_{1:T} \tag{17}$$

$$= \log \mathbb{E}_{p(X_{1:T}|X_0, \mathbf{C})}\left[\frac{p_\theta(X_{0:T}|\mathbf{C})}{p(X_{1:T}|X_0, \mathbf{C})}\right] \tag{18}$$

$$\geq \mathbb{E}_{p(X_{1:T}|X_0, \mathbf{C})}\left[\log \frac{p_\theta(X_{0:T}|\mathbf{C})}{p(X_{1:T}|X_0, \mathbf{C})}\right] \tag{19}$$

By Simplification, Eq. 19 can be reduced to the following:

$$\arg\max_\theta \mathbb{E}_{q(X_{1:T}|X_0, \mathbf{C})}\left[\log \frac{p_\theta(X_{0:T}|\mathbf{C})}{p(X_{1:T}|X_0, \mathbf{C})}\right] \Leftrightarrow \arg\min_\theta \frac{1}{2\sigma_t^2}\frac{\bar{\alpha}_{t-1}(1 - \alpha_t)^2}{(1 - \bar{\alpha}_t)^2}\|\hat{X}_\theta(X_t, t, \mathbf{C}) - X_0\|_2^2 \tag{20}$$

where $\sigma_t = \frac{1 - \bar{\alpha}_{t-1}}{1 - \bar{\alpha}_t}\beta_t$. After removing constant terms, we obtain the denoising loss in diffusion models:

$$L_{\text{hoi\_diff}}(\hat{X}_\theta, \mathbf{C}) = \mathbb{E}_{X_0 \sim p(X_0), X_t \sim p(X_t|X_0), t \sim [1,T]}\|X_0 - X_\theta(X_t, t, \mathbf{C})\|_2^2 \tag{21}$$

We also introduce geometric loss, including distance map loss $L_{\text{hoi\_distance}}$ and relative orientation loss $L_{\text{hoi\_orient}}$ for physical plausibility. To enable classifier-free guidance, we randomly mask 10% of the condition to train an unconditional model $X_\theta(X_t, t, \emptyset)$. Since the unconditional model captures the natural HOI sequence, it is then utilized as a prior to generate seamless transitions between different HOI sequences.

**Sampling Process.** During sampling, we employ classifier-free guidance to enhance alignment with the conditioning input $\mathbf{C}$. This approach demonstrates superior performance compared to using only the conditional model $X_\theta(X_t, t, \mathbf{C})$:

$$X_\theta^s(X_t, t, \mathbf{C}) = X_\theta(X_t, t, \emptyset) + s \cdot (X_\theta(X_t, t, \mathbf{C}) - X_\theta(X_t, t, \emptyset)), \tag{22}$$

where $s \geq 1$ controls the guidance strength. We generate samples through an iterative denoising process using the reverse diffusion posterior:

$$p(X_{t-1}|X_0, X_t) = \frac{q(X_t|X_{t-1}, X_0)q(X_{t-1}|X_0)}{q(X_t|X_0)} \tag{23}$$

$$= \frac{\mathcal{N}\left(X_t; \sqrt{\alpha_t}X_{t-1}, (1-\alpha_t)\mathbf{I}\right)\mathcal{N}\left(X_{t-1}; \sqrt{\bar{\alpha}_{t-1}}X_0, (1-\bar{\alpha}_{t-1})\mathbf{I}\right)}{\mathcal{N}\left(X_t; \sqrt{\bar{\alpha}_t}X_0, (1-\bar{\alpha}_t)\mathbf{I}\right)} \tag{24}$$

$$= \mathcal{N}\left(X_{t-1}; \frac{\sqrt{\alpha_t}(1-\bar{\alpha}_{t-1})X_t + \sqrt{\bar{\alpha}_{t-1}}(1-\alpha_t)X_0}{1-\bar{\alpha}_t}, \frac{(1-\alpha_t)(1-\bar{\alpha}_{t-1})}{1-\bar{\alpha}_t}\mathbf{I}\right) \tag{25}$$

$$\approx \mathcal{N}\left(X_{t-1}; \frac{\sqrt{\alpha_t}(1-\bar{\alpha}_{t-1})X_t + \sqrt{\bar{\alpha}_{t-1}}(1-\alpha_t)X_\theta^s(X_t, t, \mathbf{C})}{1-\bar{\alpha}_t}, \frac{(1-\alpha_t)(1-\bar{\alpha}_{t-1})}{1-\bar{\alpha}_t}\mathbf{I}\right) \tag{26}$$

### B.4 Loss-guided Physical Refinement

Loss guidance is a technique that minimizes the off-the-shelf loss function $L(X_0, y)$ during the sampling time:

$$\min_{X_0} L(X_0, y)$$
$$\text{s.t. } X_0 \in \mathcal{M} \tag{27}$$

where $y$ is the conditioning input, $\mathcal{M}$ denotes the conditional data manifold that follows the conditional distribution $p(X_0|\mathbf{C})$ learned by the diffusion model. In this work, we propose a novel loss-guided sampling strategy that explicitly enforces physical constraints during the denoising process to achieve more realistic hand-object interactions.

## C   Additional Experiments

### C.1   Results on Extreme-Case: Completely Unseen Datasets

**Result on H2O.**   We further subject our model to extreme-case testing to stress its generalization under the most challenging conditions. All objects and instructions in the **H2O** dataset are **entirely novel** to models trained on GRAB and ARCTIC, making evaluation on H2O a particularly stringent test of generalization. Despite this extreme distribution shift, our experiments demonstrate that the proposed model nonetheless delivers **robust** and **state-of-the-art** performance (shown in Table A1).

Table A1: **Unseen Results on H2O.** Training on GRAB / ARCTIC and evaluation on H2O.

| | Method | MPJPE↓ | FOL↓ | FID↓ | Diversity→ | MModality↑ |
|---|---|---|---|---|---|---|
| | GT | – | – | – | 3.43 | – |
| GRAB | MDM[39] | 95.12±3.21 | 0.64±0.04 | 70.54±2.75 | 2.36±0.11 | 12.50±0.50 |
| | TM2T[10] | 90.45±4.50 | 0.68±0.03 | 65.47±3.20 | 2.51±0.10 | 14.45±0.60 |
| | MotionGPT[16] | 85.38±4.00 | 0.61±0.02 | 60.12±2.80 | 2.73±0.13 | 15.93±0.70 |
| | Text2HOI[2] | 80.25±3.80 | 0.63±0.025 | 55.23±2.50 | 1.90±0.14 | 17.00±0.80 |
| | Ours | **75.78±4.68** | **0.52±0.49** | **51.33±3.41** | **3.07±0.27** | **18.15±1.48** |
| ARCTIC | MDM[39] | 105.32±5.00 | 0.80±0.04 | 75.89±3.50 | 1.90±0.09 | 11.27±0.35 |
| | TM2T[10] | 98.47±4.80 | 0.82±0.03 | 72.55±3.30 | 2.10±0.10 | 13.05±0.45 |
| | MotionGPT[16] | 93.15±4.50 | 0.74±0.02 | 65.37±3.00 | 2.30±0.11 | 12.96±0.55 |
| | Text2HOI[2] | 88.02±4.30 | 0.71±0.025 | 60.28±2.80 | 1.50±0.12 | 14.78±0.65 |
| | Ours | **81.36±5.77** | **0.63±0.12** | **55.78±3.62** | **2.69±0.43** | **15.44±1.48** |

### C.2   Evaluation for physical realism

We supplemented our experiments by evaluating Physical Realism and IV metrics against the closest baseline, Text2HOI (HOIGPT's code is not publicly available) in A2, and conducted ablation studies on our Physical Refinement module in A3.

Table A2: **Comparison with Text2HOI**

| Method | Physical realism ↑ | IV ↓ |
|---|---|---|
| **Seen** | | |
| Text2HOI | 0.87±0.03 | 11.74±1.22 |
| Ours | **0.93±0.02** | **9.25±0.73** |
| **Unseen** | | |
| Text2HOI | 0.79±0.05 | 14.63±1.07 |
| Ours | **0.89±0.01** | **10.35±0.82** |

Table A3: **Ablation Study on Physical Refinement**

| Method | Physical realism ↑ | IV ↓ |
|---|---|---|
| **Seen** | | |
| w/o Physical Refinement | 0.89±0.07 | 10.75±0.80 |
| Ours | **0.93±0.02** | **9.25±0.73** |
| **Unseen** | | |
| w/o Physical Refinement | 0.84±0.03 | 12.27±0.48 |
| Ours | **0.89±0.01** | **10.35±0.82** |

## C.3 3D MLLM Fine-tuning

We fine-tune the MLLM on the Affordance dataset [52] and the HOI dataset [38, 6], We first perform coarse-grained fine-tuning on the Affordance dataset to instill strong affordance priors, and then carry out fine-grained tuning on the HOI dataset to produce our final model [43]. The results shown in Table. A4.

Table A4: **MLLM Coarse-to-Fine Affordance Tuning**

| Method | AUC ↑ |
|---|---|
| w/o Fine-tuning | 68.77 |
| Coarse-grained tuning | 84.65 |
| Coarse-to-Fine Tuning (full model) | 87.02 |

## C.4 Sensitivity Analysis

We conducted a sensitivity analysis on the guidance rate, and the results are as follows(shown in Table A5 and Table A6). Our experimental results demonstrate that the proposed model maintains robust performance even under these challenging conditions.

## C.5 Ablation Study on Multi <AFF>

The additional ablation study results are as follows A7.

## C.6 Ablation Study on Motion In-between Refinement

**Motion In-between Metric.** To the best of our knowledge, no previous work has defined an evaluation metric for motion in-between hand-object interaction. We thus introduce a simple yet effective measure, the "Smooth Rate," to quantify the temporal continuity of interpolated motion segments as follows,

$$\text{SmoothRate} = \frac{\mathrm{dFID}}{\mathrm{d}t}, \tag{28}$$

where $\mathrm{d}t$ is the derivative of time. The results are shown in Table A8 and Table A9.

Table A5: **Guidance Rate on GRAB**

|  | Guidance Rate | MPJPE↓ | FOL↓ | FID ↓ | Diversity → | MModality ↑ |
|---|---|---|---|---|---|---|
|  | GT | - | - | - | 4.66 | - |
| Seen | 0.5 | 58.08±0.87 | 0.38±0.01 | 34.40±0.57 | 3.35±0.06 | 18.21±0.31 |
| Seen | 2.0 | 51.86±0.62 | 0.29±0.03 | 27.45±1.13 | 3.63±0.02 | 23.35±0.46 |
| Seen | 2.5 | **47.64±1.03** | **0.26±0.02** | **26.43±0.77** | **3.69±0.27** | **24.59±2.01** |
| Seen | 3.0 | 50.81±1.07 | 0.32±0.03 | 26.75±0.30 | 3.57±0.10 | 23.86±0.77 |
| Seen | 5.0 | 58.92±1.28 | 0.33±0.02 | 34.29±0.47 | 3.40±0.08 | 23.55±0.51 |
| Unseen | 0.5 | 61.39±2.45 | 0.40±0.02 | 35.44±1.45 | 3.32±0.11 | 13.34±0.47 |
| Unseen | 2.0 | 54.95±1.30 | 0.30±0.06 | 29.21±2.15 | 3.55±0.07 | 18.70±0.79 |
| Unseen | 2.5 | **51.34±0.85** | **0.27±0.01** | **28.29±0.62** | **3.61±0.09** | **19.91±0.63** |
| Unseen | 3.0 | 54.62±1.61 | 0.33±0.01 | 28.61±0.81 | 3.50±0.33 | 19.16±1.59 |
| Unseen | 5.0 | 62.98±1.93 | 0.35±0.04 | 37.25±0.82 | 3.34±0.14 | 18.81±1.41 |

Table A6: **Guidance Rate on ARCTIC**

|  | Guidance Rate | MPJPE↓ | FOL↓ | FID ↓ | Diversity → | MModality ↑ |
|---|---|---|---|---|---|---|
|  | GT | - | - | - | 3.39 | - |
| Seen | 0.5 | 52.23±1.06 | 0.40±0.01 | 31.05±1.56 | 1.97±0.19 | 12.77±0.45 |
| Seen | 2.0 | 46.04±1.19 | 0.28±0.03 | 20.98±2.30 | 2.62±0.04 | 14.96±0.61 |
| Seen | 2.5 | **45.15±0.94** | **0.25±0.04** | **19.74±0.16** | **2.65±0.03** | **15.25±1.44** |
| Seen | 3.0 | 46.55±1.74 | 0.27±0.02 | 21.03±0.67 | **2.68±0.15** | 15.03±1.75 |
| Seen | 5.0 | 53.25±2.07 | 0.38±0.02 | 32.85±2.04 | 3.40±0.06 | 12.86±2.07 |
| Unseen | 0.5 | 51.67±1.14 | 0.35±0.02 | 31.54±0.68 | 2.07±0.05 | 10.18±0.26 |
| Unseen | 2.0 | 47.70±0.88 | 0.30±0.02 | 20.81±2.12 | 2.46±0.07 | 12.36±0.89 |
| Unseen | 2.5 | **47.25±0.39** | **0.28±0.03** | **20.05±0.80** | **2.49±0.08** | **12.66±0.71** |
| Unseen | 3.0 | 47.76±0.66 | 0.29±0.01 | 21.07±0.41 | 2.51±0.28 | 12.50±1.77 |
| Unseen | 5.0 | 54.19±0.89 | 0.34±0.02 | 27.32±0.56 | 2.21±0.06 | 10.53±0.69 |

To determine the most suitable window size [1] for the motion in-between algorithm, we conducted the following comparative experimentsA10.

## C.7   Visualization on Affordance

In this subsection, we present visualizations of open-world affordances on seen and unseen objects in Fig. A1.

## C.8   Qualitative results Compare with SOTA

This section presents additional visual comparisons between our approach and existing state-of-the-art (SOTA) methods.

**Seen Objects.**   Qualitative results on seen objects in Fig. A2.

**Unseen Objects.**   Qualitative results on unseen objects in Fig. A3.

## C.9   Statistically Insignificant

For the ablation study, we performed paired two-sample t-tests, repeating each test five times and reporting the mean p-value. As summarized in Table A11, the proposed method is significant at the 95% confidence level for the majority of metrics(P-value ≤0.05).

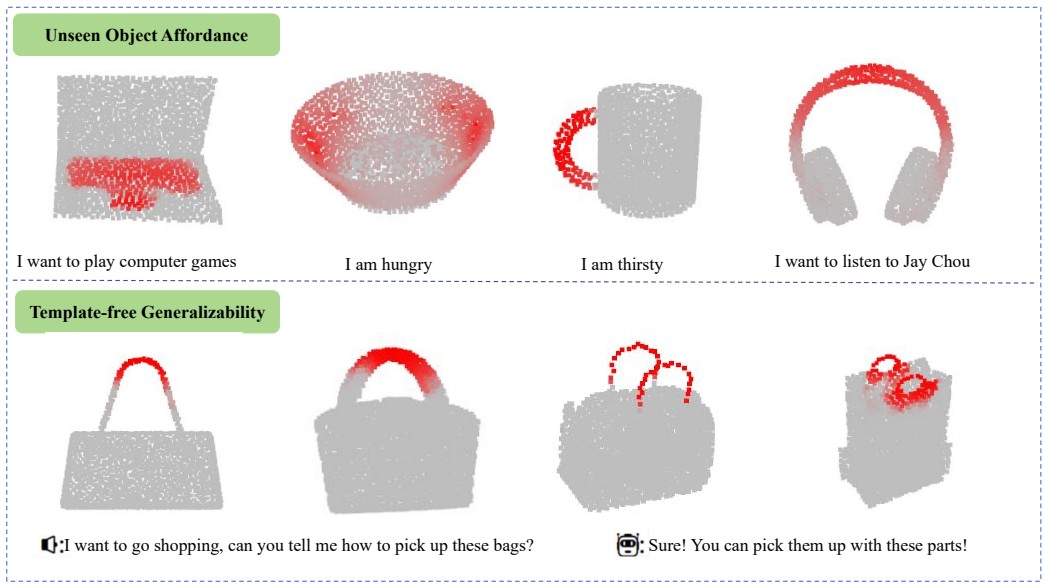

Figure A1: Visualization on Affordance

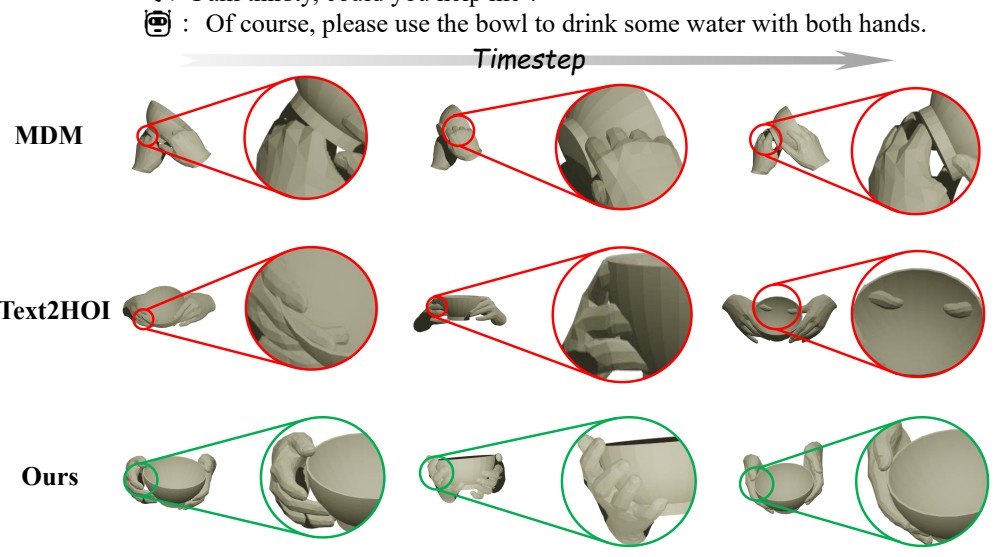

Figure A2: Qualitative results on seen object

Table A7: **Ablation on Affordance Configuration (Single vs. Multi)**

| Method | MPJPE ↓ | FOL ↓ | FID ↓ | Diversity → | MModality ↑ |
|---|---|---|---|---|---|
| **Seen** | | | | | |
| Single <AFF> | 52.05±0.82 | 0.33±0.01 | 29.31±0.76 | 3.50±0.12 | 21.05±1.47 |
| Multi <AFF> | **47.64±1.03** | **0.26±0.02** | **26.43±0.77** | **3.69±0.27** | **24.59±2.01** |
| **Unseen** | | | | | |
| Single <AFF> | 56.48±1.06 | 0.39±0.02 | 34.15±1.08 | 3.40±0.22 | 17.03±1.25 |
| Multi <AFF> | **51.34±0.85** | **0.27±0.01** | **28.29±0.62** | **3.61±0.09** | **19.91±0.63** |

Table A8: **Ablation Study of Motion In-between Results on GRAB**

| Setting | Method | SmoothRate ↓ |
|---|---|---|
| Seen | w/o Motion In-between | 38.18 ± 6.75 |
| | **Ours** | **2.98 ± 0.43** |
| Unseen | w/o Motion In-between | 35.25 ± 4.91 |
| | **Ours** | **3.70 ± 0.61** |

# D   Code and Dataset

## D.1   Code

We will release our code as soon as possible. GitHub is OpenHOI

## D.2   Dataset

**H2O.**   This dataset contains 571,645 synchronized multi-view RGB-D frames captured with five Kinect sensors in three indoor scenes. Each frame includes 3D poses for both hands, 6-DoF object poses, and verb–noun action labels (36 classes). Split into training (344,645), validation (73,380), and test (153,620) frames, H2O supports egocentric interaction recognition and manipulation benchmarks.

🔊: I want to read that interesting story again!
🤖 :   Reading a storybook is a great choice. Please cradle the volume with one hand on each cover and bring it to a comfortable viewing angle and distance.

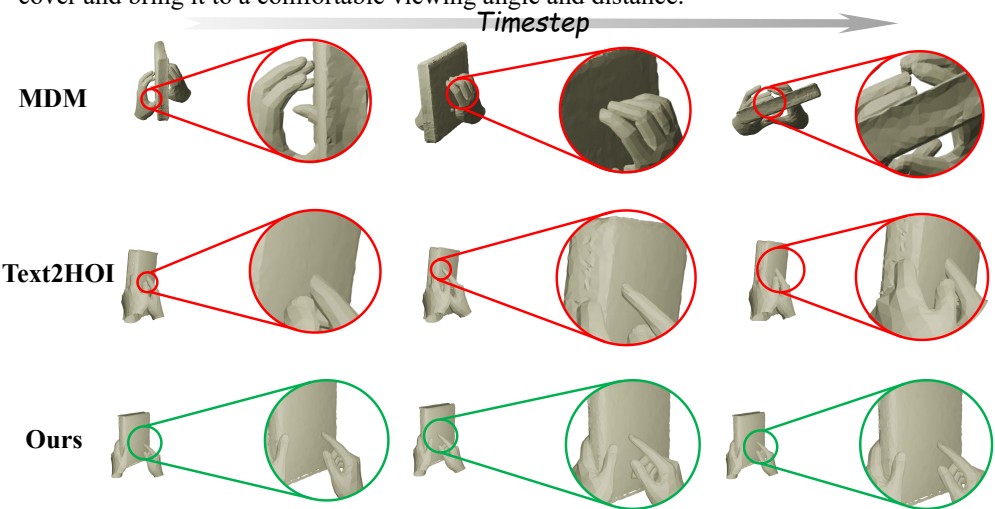

Figure A3: Qualitative results on unseen object

Table A9: **Ablation Study of Motion In-between Results on ARCTIC**

| Setting | Method | SmoothRate ↓ |
|---------|--------|--------------|
| Seen | w/o Motion In-between | $41.37 \pm 5.98$ |
|  | **Ours** | $\mathbf{6.77 \pm 2.17}$ |
| Unseen | w/o Motion In-between | $44.05 \pm 5.63$ |
|  | **Ours** | $\mathbf{6.04 \pm 3.25}$ |

Table A10: **Window Size Compare on GRAB**

| Size | SmoothRate (Seen) ↓ | SmoothRate (Unseen) ↓ |
|------|---------------------|------------------------|
| 1 | $4.77 \pm 0.68$ | $5.26 \pm 2.47$ |
| 3 | $3.59 \pm 0.81$ | $4.58 \pm 1.05$ |
| 5 | $\mathbf{2.98 \pm 0.43}$ | $\mathbf{3.70 \pm 0.61}$ |
| 10 | $3.24 \pm 0.65$ | $4.08 \pm 0.82$ |
| 20 | $3.30 \pm 0.57$ | $4.19 \pm 0.74$ |

**Dataset annotation.** For both the GRAB [38] and ARCTIC [6] datasets, we preprocess the datasets and annotate the semantics. After that, the object point clouds are first upsampled to gain accurate affordance maps using our inference model while ensuring fine geometric details in the meantime. Our model then processes the upsampled data to infer accurate affordance maps, which are subsequently downsampled to match the original resolution for efficient computation.

We preprocess both datasets and get their semantic annotations. First, we upsample the object point clouds to enhance geometric details and generate accurate affordance maps using our inference model. The upsampled data is then processed to infer affordance maps, which are downsampled back to the original resolution for computational efficiency.

To enhance the semantic alignment between language and interaction, we employ a large language model (LLM) to refine the original hand-object interaction (HOI) descriptions. The LLM generates high-level, natural language annotations that better capture the intent and dynamics of HOI.

## E  Discussion

### E.1  Generalizability ability in HOI: Affordance as a key

Affordance is a powerful and explicit prior for interaction that can guide complex and fine-grained HOI synthesis. Our model achieves strong generalization by using affordance as a middleware layer.

- Open-World Affordance Grounding: We first employ a coarse-to-fine tuning strategy to equip the model with strong affordance reasoning capabilities, enabling it to generate open-world affordance grounding. This enables the synthesis of realistic HOI sequences, demonstrating strong template-free generalization capabilities.

- Affordance serves as a crucial condition: The open-world affordance grounding serves as a crucial condition for the affordance-driven HOI Diffusion.We incorporate affordance not only during training but also in the loss-guidance applied during inference.

- Affordance-based Refinement: we design an improved refinement strategy based on affordance: for the interaction between the hand and the object, we optimize based on affordance grounding rather than the conventional closest-surface-point approach.

- A Template-free Example: Our 3D MLLM has learned from a wide variety of cups, it can still generate accurate affordance grounding for a completely unseen mug. Accurate affordance grounding will guide the synthesis of realistic HOI sequences.

Table A11: **Two-test Statistically Insignificant**

| Metric | w/o Affordance | w/o CFG | w/o $l_{\text{penetration}}$ | w/o $l_{\text{aff}}$ |
|---|---|---|---|---|
| MPJPE (Seen) | $3.1 \times 10^{-6}$ | $2.0 \times 10^{-4}$ | $7.5 \times 10^{-5}$ | $2.6 \times 10^{-2}$ |
| MPJPE (Unseen) | $4.4 \times 10^{-5}$ | $1.2 \times 10^{-4}$ | $5.5 \times 10^{-4}$ | $2.6 \times 10^{-4}$ |
| FOL (Seen) | $1.7 \times 10^{-3}$ | $6.8 \times 10^{-5}$ | $1.7 \times 10^{-5}$ | $3.1 \times 10^{-5}$ |
| FOL (Unseen) | $2.2 \times 10^{-2}$ | $2.0 \times 10^{-3}$ | $4.9 \times 10^{-2}$ | $3.0 \times 10^{-1}$ |
| FID (Seen) | $2.4 \times 10^{-5}$ | $1.5 \times 10^{-3}$ | $1.0 \times 10^{-2}$ | $2.0 \times 10^{-3}$ |
| FID (Unseen) | $2.2 \times 10^{-5}$ | $8.7 \times 10^{-5}$ | $2.0 \times 10^{-4}$ | $2.5 \times 10^{-4}$ |
| Diversity (Seen) | $1.5 \times 10^{-1}$ | $1.0 \times 10^{-1}$ | $4.4 \times 10^{-2}$ | $1.4 \times 10^{-1}$ |
| Diversity (Unseen) | $1.8 \times 10^{-3}$ | $3.0 \times 10^{-4}$ | $4.6 \times 10^{-2}$ | $4.8 \times 10^{-2}$ |
| MModality (Seen) | $2.6 \times 10^{-4}$ | $2.5 \times 10^{-2}$ | $5.0 \times 10^{-3}$ | $1.1 \times 10^{-2}$ |
| MModality (Unseen) | $1.0 \times 10^{-3}$ | $4.7 \times 10^{-2}$ | $2.2 \times 10^{-2}$ | $2.2 \times 10^{-2}$ |

## E.2 How to use 3D MLLM in Emboided AI: Choose Powerful Foundation Model and Coarse-to-fine tuning

The 3D multimodal large model has been widely applied in HOI and embodied intelligence, and it is very important to choose a basic model suitable for downstream tasks. In OpenHOI, we chose ShapeLLM as our base model, ShapeLLM is a powerful 3D foundation model that performs exceptionally well on multiple downstream tasks (e.g., Embodied Visual Grounding, Visual Question Answering, and Scene Understanding), making it highly suitable for HOI tasks.After research, we found that ShapeLLM has the following advantages and disadvantages

**Advantages:** ShapeLLM is trained on a large amount of 3D embodied interaction data and achieves state-of-the-art performance across various downstream tasks. It possesses strong priors in 3D interaction and demonstrates impressive zero-shot 3D representation capabilities.

**Disadvantages:** ShapeLLM has not been trained on part-level object annotations, which limits its reasoning capabilities for fine-grained object understanding.

In order to make the selected 3D base model as suitable as possible for our task, we need to use data to fine tune the model. We adopt a coarse-to-fine tuning strategy: we first pre-train the model on an object-centric affordance dataset to enable it to acquire strong affordance priors. Then, we fine-tune the model on HOI datasets to better align the semantics with the target domain. This allows the model to learn highly effective affordance representations.

## E.3 Future of Work: Real-World Applications, about AR/VR and Robotics

OpenHOI can be extended to a wide range of future work in other fields. We have listed several noteworthy areas and provided preliminary solutions for the challenges in future applications

**Robotics Manipularion:** OpenHOI can be integrated into real-world robotic manipulation systems, including industrial robot arms and service robots, to enable more flexible and human-like interactions[48, 54, 55].

- Open-World Affordance Grounding as Powerful Guidance for Robots: Leveraging OpenHOI's 3D MLLM, our method performs open-world affordance grounding to facilitate the identification of feasible grasping, pushing, and tool-use regions on novel objects, thereby significantly improving success rates for pick-and-place, assembly, and tool-handling tasks.

- Realistic HOI sequences synthesis for Robot Manipulation: The HOI sequences generated by OpenHOI can be adapted into robotic manipulation sequences. First, we can use an extraction algorithm to obtain the object's 6-DoF pose. Then, inverse kinematics are applied to the wrist parameters to compute the robot arm's pose. Finally, a retargeting algorithm transfers the human hand motions onto various robotic hand configurations for manipulation. This approach ensures smooth, precise, and robust manipulation behaviors in real-world deployments.

**Virtual Reality Vision:** By synthesizing realistic 3D hand–object interaction sequences, OpenHOI enables users to manipulate virtual objects naturally, for example, by picking up, twisting, or pouring items, thereby enhancing immersion in training simulators, gaming, and virtual prototyping.

**Challenges:** Robotic manipulation tasks typically demand rapid inference. We plan to employ DPM-Solver to accelerate the diffusion inference process, which is an important direction for our future work[31].

