# OpenReview forum: "OpenHOI: Open-World Hand-Object Interaction Synthesis with Multimodal Large Language Model"
_NeurIPS.cc/2025/Conference — NeurIPS 2025 oral_

### Official Review · Reviewer_juyL · 2025-06-17

**Clarity:** 3
**Significance:** 3
**Originality:** 3
**Rating:** 4
**Confidence:** 2

**Summary:**

The paper proposes a method for 3D hand-object interactions leveraging an MLLM. The use of MLLM allows for long-horizon manipulation. The framework leverages a pre-trained MLLM and diffusion model, and minimal training using LoRA adapters. In addition, several refinements during and post-training are applied to account for the geometrical shortcomings of LLMs.

**Questions:**

The paper is well written. Most questions are regarding error analysis, as discussed more extensively above, but the key things to address are:

- Clarity on the \<aff\> tokens is necessary
- Some results appear statistically insignificant; it would be best to show the importance of each component
- Failure cases, error analysis and limitations are missing

**Ethical Concerns:**

["NO or VERY MINOR ethics concerns only"]

**Final Justification:**

The authors have addressed most of my and other reviewers' concerns. As such, I will maintain my original positive rating.
The only thing that remains a bit of a concern is the claim for long-horizon manipulation; in the rebuttal, the authors admit performance degrading after three actions, which does not seem very long.

**Limitations:**

The authors have not discussed limitations and societal impact, both of which should be included in the paper.

**Paper Formatting Concerns:**

Diversity arrows on tables seem incorrect.

**Quality:**

3

**Strengths And Weaknesses:**

I have little experience in the HOI task (which is reflected in the confidence of this review). As such, the comments are primarily regarding the MLLM and the completeness of the experiment section.
- is the method using a single \<Aff\> token? it appears as if multiple are used in Fig 2, but only one is mentioned in the methodology section. If multiple are used, these need to be made explicit in the method section, and relevant experiments should be included in the ablation studies.
- Furthermore, all results presented would benefit from statistical significance testing, as mean and standard deviation are provided. This is important, as some results in the ablation (eg, MPJPE and Diversity) appear comparable, and the importance of each component may be undermined (eg, affordance loss).
- The method should include failure cases and an error analysis.
- How long is the horizon before performance deteriorates? What instructions/samples does the method fail to address?
- Are there any limitations to the method?

---

> ### Author Rebuttal · Authors · 2025-07-31
>
> We sincerely value Reviewer juyL's comments, which are helpful. A point-by-point reply to your comments is provided below.
>
>
> >Q1：Is the method using a single \<Aff\> token? It appears as if multiple are used in Fig 2, but only one is mentioned in the methodology section. If multiple are used, these need to be made explicit in the method section, and relevant experiments should be included in the ablation studies.
>
> A1: Yes, our method uses multi \<AFF\> tokens. Lines 139-141 and Formula (6) show that we use **S** occurrences of the segmentation token \<AFF\>. The additional ablation study results are as follows:
>
>
> | Method|MPJPE $\downarrow$|FOL $\downarrow$|FID $\downarrow$|Diversity $\rightarrow$|MModality $\uparrow$|
> |:------------:|:-----:|:----:|:----:|:---------:|:---------:|
> |Single \<AFF\>(Seen)|52.05±0.82|0.33±0.01|29.31±0.76|3.50±0.12|21.05±1.47|
> |Multi \<AFF\>(Seen)| **47.64±1.03**|**0.26±0.02**|**26.43±0.77**|**3.69±0.27**|**24.59±2.01**|
> |Single \<AFF\>(Unseen)|56.48±1.06|0.39±0.02|34.15±1.08|3.40±0.22|17.03±1.25|
> |Multi \<AFF\>(Unseen)|**51.34±0.85**|**0.27±0.01**|**28.29±0.62**|**3.61±0.09**|**19.91±0.63**|
>
>
>
> >Q2: Furthermore, all results presented would benefit from statistical significance testing, as mean and standard deviation are provided. This is important, as some results in the ablation (eg, MPJPE and Diversity) appear comparable, and the importance of each component may be undermined (eg, affordance loss).
>
> A2: Thank you for your suggestion. For the ablation study, we performed paired two‑sample t‑tests, repeating each test five times and reporting the mean p‑value. As summarized in Table , the proposed method is significant at the 95% confidence level for the majority of metrics (P-value $\leqslant$ 0.05).
>
> | Metric               | w/o Affordance | w/o CFG     | w/o $l_{penetration}$ | w/o $l_{aff}$    |
> |:---------------------:|:---------------:|:------------:|:------------------:|:-------------:|
> | MPJPE (Seen)         | 3.1×10⁻⁶       | 2.0×10⁻⁴    | 7.5×10⁻⁵          | 2.6×10⁻²     |
> | MPJPE (Unseen)       | 4.4×10⁻⁵       | 1.2×10⁻⁴    | 5.5×10⁻⁴          | 2.6×10⁻⁴     |
> | FOL (Seen)           | 1.7×10⁻³       | 6.8×10⁻⁵    | 1.7×10⁻⁵          | 3.1×10⁻⁵     |
> | FOL (Unseen)         | 2.2×10⁻²       | 2.0×10⁻³    | 4.9×10⁻²          | 3.0×10⁻¹     |
> | FID (Seen)           | 2.4×10⁻⁵       | 1.5×10⁻³    | 1.0×10⁻²          | 2.0×10⁻³     |
> | FID (Unseen)         | 2.2×10⁻⁵       | 8.7×10⁻⁵    | 2.0×10⁻⁴          | 2.5×10⁻⁴     |
> | Diversity (Seen)     | 1.5×10⁻¹       | 1.0×10⁻¹    | 4.4×10⁻²          | 1.4×10⁻¹     |
> | Diversity (Unseen)   | 1.8×10⁻³       | 3.0×10⁻⁴    | 4.6×10⁻²          | 4.8×10⁻²     |
> | MModality (Seen)     | 2.6×10⁻⁴       | 2.5×10⁻²    | 5.0×10⁻³          | 1.1×10⁻²     |
> | MModality (Unseen)   | 1.0×10⁻³       | 4.7×10⁻²    | 2.2×10⁻²          | 2.2×10⁻²     |
>
>
>
> >Q3: Failure cases, error analysis and limitations are missing
>
> A3: Thanks for your question. We enumerate the failure cases, error analysis and limitations below.
>
> **Failure Cases and Error Analysis:** Given a scenario, there is a row of cabinets, and the instruction "Open the cabinet", our model can open one but cannot target a specific one (e.g., "Open the second cabinet"). This limitation arises because the model has not been trained on a large-scale 3D QA dataset, resulting in reduced logical reasoning capabilities.
>
> **Limitations:** As stated in the Conclusion section, we have outlined our limitations and future work (lines 292-297). We acknowledge that our model currently struggles with (1) highly complex, scene‑level, long‑horizon interactions (e.g., "cooking a meal") and (2) extremely fine‑grained dynamics on articulated objects (e.g., "fluid simulation in pouring tasks"). Addressing these challenges will be the focus of our future work.
>
>
>
> >Q4: How long is the horizon before performance deteriorates?
>
> A4: Due to the accumulation of model errors, performance typically degrades after more than three consecutive actions.
>
>
>
> >Q5: Diversity arrows on tables seem incorrect.
>
> A5: Our diversity arrows on tables are correct. In fact, in the HOI domain, sequence diversity is deemed better the more closely it matches the ground‑truth distribution of the original dataset; this is exactly how Text2HOI and HOIGPT report their results.

---

> > ### Comment · Reviewer_juyL · 2025-08-05
> >
> > Thank you for the detailed rebuttal! Most of my concerns regarding experiments have been addressed, so I will take this into consideration for the final score.

---

> > > ### Author Response · Authors · 2025-08-05
> > >
> > > Thank you for your thorough review comments. We are glad that our responses have addressed most of your concerns. We will make our experiments more comprehensive in the final version of the paper.

---

### Official Review · Reviewer_wJWx · 2025-06-29

**Clarity:** 4
**Significance:** 4
**Originality:** 4
**Rating:** 5
**Confidence:** 5

**Summary:**

The paper proposes OpenHOI, the first open-world hand-object interaction (HOI) synthesis framework capable of generating long-horizon manipulation sequences for novel objects guided by open-vocabulary instructions. The paper's contributions are significant and address a challenging problem in the field of 3D interaction research. The proposed approach combines a 3D multimodal large language model (MLLM) with an affordance-driven diffusion model and physics-aware refinement, achieving strong generalization capabilities across object categories, task horizons, and linguistic complexity. Experimental results demonstrate the superiority of the method over existing state-of-the-art approaches.

**Questions:**

1. The paper adopts ShapeLLM as the backbone of the 3D MLLM. What are the advantages and disadvantages of this choice compared to the other 3D MLLM models? How does ShapeLLM's performance in the context of OpenHOI align with its original design goals and capabilities?
2. What are the potential applications of the proposed OpenHOI framework in real-world scenarios, such as robotics and virtual reality? What challenges might arise in practical applications, and how can they be addressed?

**Ethical Concerns:**

["NO or VERY MINOR ethics concerns only"]

**Final Justification:**

After reading the rebuttal, I keep my acceptance score for this paper.

**Limitations:**

Please find in weakness and questions.

**Paper Formatting Concerns:**

This paper does not have formatting issues.

**Quality:**

4

**Strengths And Weaknesses:**

Strengths
1. The paper presents the first open-world hand-object interaction (HOI) synthesis framework, OpenHOI, which can generate long-horizon manipulation sequences for novel objects guided by open-vocabulary instructions. This represents a significant advancement in the field of 3D interaction research, addressing a previously unresolved challenge.
2. The proposed method demonstrates strong generalization capabilities across object categories, task horizons, and linguistic complexity. It outperforms existing state-of-the-art methods in handling unseen objects and complex, linguistically diverse commands.
3. The paper is well-organized and clearly written. The authors provide a detailed introduction to the research background and motivation, comprehensively review related work, meticulously describe the proposed method, and thoroughly evaluate the experimental results.
Weakness
1. Hierarchical Task Decomposition Limitations: The paper notes that while OpenHOI supports multi-stage tasks, it struggles with compositional long-horizon sequences like "cook a meal" due to limitations in hierarchical task decomposition. This suggests that the method may not be suitable for highly complex and hierarchical tasks.
2. Limitations in Fine-Grained Dynamics: Although the physics-aware refinement improves interaction plausibility, the paper acknowledges challenges in handling fine-grained dynamics, such as fluid simulation for pouring tasks. This indicates that the proposed method may not fully capture the complexities of certain interactions.

---

> ### Author Rebuttal · Authors · 2025-07-31
>
> We sincerely thank Reviewer wJWx for his/her comments and questions. We offer point-by-point clarifications to the issues you raised.
>
> > Q1-1: The paper adopts ShapeLLM as the backbone of the 3D MLLM. What are the advantages and disadvantages of this choice compared to the other 3D MLLM models?
>
> A1-1：Thank you for your question. We enumerate the advantages and disadvantages below.
>
> **Advantages:** ShapeLLM is trained on a large amount of 3D embodied interaction data and achieves state-of-the-art performance across various downstream tasks. It possesses strong priors in 3D interaction and demonstrates impressive zero-shot 3D representation capabilities.
>
> **Disadvantages:** ShapeLLM has not been trained on part‑level object annotations, which limits its reasoning capabilities for fine‑grained object understanding.
>
>
>
> >Q1-2: How does ShapeLLM's performance in the context of OpenHOI align with its original design goals and capabilities?
>
> A1-2：First, ShapeLLM is a powerful 3D foundation model that performs exceptionally well on multiple downstream tasks (e.g., Embodied Visual Grounding, Visual Question Answering, and Scene Understanding), making it highly suitable for HOI tasks.
>
> Second, we adopt a coarse-to-fine tuning strategy: we first pre-train the model on an object-centric affordance dataset to enable it to acquire strong affordance priors. Then, we fine-tune the model on HOI datasets to better align the semantics with the target domain. This allows the model to learn highly effective affordance representations.
>
>
>
> >Q2: What are the potential applications of the proposed OpenHOI framework in real-world scenarios, such as robotics and virtual reality? What challenges might arise in practical applications, and how can they be addressed?
>
>
> A2: Thank you for your suggestion. Applying OpenHOI to real‑world scenarios is both a challenging and compelling endeavor.
>
> **Robotics Application:** OpenHOI can be integrated into real‑world robotic manipulation systems, including industrial robot arms and service robots, to enable more flexible and human‑like interactions.
> - **Open-World Affordance Grounding as Powerful Guidance for Robots**:Leveraging OpenHOI's 3D MLLM, our method performs open-world affordance grounding to facilitate the identification of feasible grasping, pushing, and tool-use regions on **novel objects**, thereby significantly improving success rates for pick-and-place, assembly, and tool-handling tasks.
> - **Realistic HOI sequences synthesis for Robot Manipulation:** The HOI sequences generated by OpenHOI can be adapted into robotic manipulation sequences. First, we can use an extraction algorithm to obtain the object's **6-DoF pose**. Then, **inverse kinematics** are applied to the wrist parameters to compute the robot arm's pose. Finally, a **retargeting** algorithm transfers the human hand motions onto various robotic hand configurations for manipulation. This approach ensures smooth, precise and robust manipulation behaviors in real‑world deployments.
>
> **Virtual Reality Application:** By synthesizing realistic 3D hand–object interaction sequences, OpenHOI enables users to **manipulate virtual objects** naturally, for example by picking up, twisting, or pouring items, thereby enhancing immersion in training simulators, gaming, and virtual prototyping.
>
>
> **Challenges:** Robotic manipulation tasks typically demand rapid inference. We plan to employ DPM‑Solver to accelerate the diffusion inference process important direction for our future work.

---

### Official Review · Reviewer_7WnN · 2025-06-30

**Clarity:** 2
**Significance:** 3
**Originality:** 3
**Rating:** 4
**Confidence:** 3

**Summary:**

This paper introduces OpenHOI, the first framework for open-world hand-object interaction (HOI) synthesis. It enables the generation of long-horizon HOI sequences for novel objects conditioned on language instructions. The core technical idea is to leverage the embedded knowledge of a 3D Multimodal Large Language Model (MLLM) to enhance the generalizability of HOI generation. Specifically, the approach fine-tunes a pre-trained 3D MLLM for affordance grounding and semantic task decomposition. These outputs are then used in a diffusion-based HOI synthesis and a physics-based refinement stage to produce the final HOI sequences. IIn experiments, OpenHOI achieves new state-of-the-art performance on HOI generation for both seen and unseen objects, outperforming existing state-of-the-art methods.

**Questions:**

Please see the Weaknesses section.

**Ethical Concerns:**

["NO or VERY MINOR ethics concerns only"]

**Final Justification:**

During the author–reviewer discussion, most of my concerns were addressed, so I have decided to maintain my positive rating.

**Limitations:**

Yes.

**Quality:**

3

**Strengths And Weaknesses:**

[Strengths]

1. Writing quality

Overall, the paper is easy to read, and the motivation and technical contributions are clearly presented.

2. Good motivation

I agree that enhancing the generalizability of HOI generation to unseen object categories is an important research direction, as this remains a known limitation of existing HOI generation methods. In this context, leveraging the embedded knowledge of a pre-trained 3D MLLM is a reasonable and promising approach.

3. Strong method novelty

The proposed method is novel and interesting. This is one of the first works to leverage a large language model for HOI generation, and the approach aligns well with the high-level goal of improving generalizability, as noted in the previous point.


[Weaknesses]

1. Questionable generalizability of the diffusion-based HOI generation module

The proposed framework consists of two sequential stages: (1) 3D MLLM-based instruction decomposition and affordance reasoning, and (2) diffusion-based HOI generation conditioned on the outputs of the 3D MLLM. While the first stage may offer high generalizability by leveraging the pre-trained 3D MLLM, it remains unclear how the second, diffusion-based module achieves strong generalizability. In the experiments, the performance gap relative to the existing state-of-the-art baseline (Text2HOI) does not noticeably differ between seen and unseen objects. This raises the question of whether the improved performance is primarily due to stronger representation capacity rather than actual improvements in generalization, which is the main claim of the paper.

2. Missing evaluation for physical realism

One of the claimed technical contributions is the introduction of a physics-based refinement stage to enhance the physical realism of the generated HOI sequences. However, empirical validation for this component is missing, despite the fact that the closest baselines (e.g., Text2HOI, HOIGPT) report evaluation metrics specifically designed to assess physical realism (e.g., physical realism, interpenetration volume).

3. Questionable performance on long-horizon generation

Another core contribution claimed by the paper is the ability to generate long-horizon HOI sequences. However, I have concerns about this claim due to the method design. Methodologically, the approach performs in-between motion refinement by enforcing smoothness only between adjacent frames (Eq. 11), which contrasts with existing long-horizon synthesis methods (e.g., [R1]) that consider much longer transition windows. Clarifying these points would be helpful.

[R1] Barquero et al., FlowMDM: Seamless Human Motion Composition with Blended Positional Encodings, CVPR 2024.

---

> ### Author Rebuttal · Authors · 2025-07-31
>
> We appreciate Reviewer 7WnN's detailed feedback and the opportunity to clarify our work. We provide comprehensive responses to your concerns in the following.
>
> >Q1: Questionable generalizability of the diffusion-based HOI generation module.The proposed framework consists of two sequential stages: (1) 3D MLLM-based instruction decomposition and affordance reasoning, and (2) diffusion-based HOI generation conditioned on the outputs of the 3D MLLM. While the first stage may offer high generalizability by leveraging the pre-trained 3D MLLM, it remains unclear how the second, diffusion-based module achieves strong generalizability. In the experiments, the performance gap relative to the existing state-of-the-art baseline (Text2HOI) does not noticeably differ between seen and unseen objects. This raises the question of whether the improved performance is primarily due to stronger representation capacity rather than actual improvements in generalization, which is the main claim of the paper.
>
> A1: Thanks for your question. Affordance is a **powerful and explicit prior** for interaction that can guide complex and fine-grained HOI synthesis. Our model achieves strong generalization by using affordance as a **middleware layer**.
> - **Open-World Affordance Grounding:** We first employ a coarse-to-fine tuning strategy to equip the model with strong **affordance reasoning capabilities**, enabling it to generate **open-world** affordance grounding. This enables the synthesis of realistic HOI sequences, demonstrating strong **template-free** generalization capabilities.
> - **Affordance serves as a crucial condition:** The open-world affordance grounding serves as a crucial condition for the affordance-driven HOI Diffusion.We incorporate affordance not only during training but also in the loss-guidance applied during inference.
> - **Affordance-based Refinement:** we design an improved refinement strategy based on **affordance**: for the interaction between the hand and the object, we optimize based on **affordance grounding** rather than the conventional closest-surface-point approach.
> - **A Template-free Example:** Our 3D MLLM has learned from a wide variety of cups, it can still generate accurate affordance grounding for a completely unseen mug. **Accurate affordance grounding** will guide the synthesis of realistic HOI sequences.
>
>
>
>
>
> >Q2: Missing evaluation for physical realism. One of the claimed technical contributions is the introduction of a physics-based refinement stage to enhance the physical realism of the generated HOI sequences. However, empirical validation for this component is missing, despite the fact that the closest baselines (e.g., Text2HOI, HOIGPT) report evaluation metrics specifically designed to assess physical realism (e.g., physical realism, interpenetration volume).
>
> A2: We supplemented our experiments by evaluating **Physical Realism** and **IV** metrics against the **closest baseline**, Text2HOI (HOIGPT’s code is not publicly available), and conducted ablation studies on our **Physical Refinement** module. We will include these two metrics in the final version of the paper.
>
>
> Compare with the closest baselines:
> | Method| Physical realism $\uparrow$| IV $\downarrow$|
> |:-------:|:-------:|:-------:|
> | Text2HOI(Seen) |0.87±0.03|11.74±1.22|
> | Ours(Seen) |**0.93±0.02**|**9.25±0.73**|
> | Text2HOI(Unseen) |0.79±0.05|14.63±1.07|
> | Ours(Unseen) |**0.89±0.01**|**10.35±0.82**|
>
> Ablation Studies on Physical Realism and IV:
> | Method| Physical realism $\uparrow$| IV $\downarrow$|
> |:-------:|:-------:|:-------:|
> | w/o Physical Refinement(Seen)|0.89±0.07|10.75±0.80|
> | Ours(Seen) |**0.93±0.02**|**9.25±0.73**|
> | w/o Physical Refinement(Unseen) |0.84±0.03|12.27±0.48|
> | Ours(Unseen) |**0.89±0.01**|**10.35±0.82**|
>
>
> >Q3: Questionable performance on long-horizon generation. Another core contribution claimed by the paper is the ability to generate long-horizon HOI sequences. However, I have concerns about this claim due to the method design. Methodologically, the approach performs in-between motion refinement by enforcing smoothness only between adjacent frames (Eq. 11), which contrasts with existing long-horizon synthesis methods (e.g., FlowMDM) that consider much longer transition windows. Clarifying these points would be helpful.
>
> A3: Thank you for your suggestion. Motivated by FlowMDM's longer transition windows, we conducted comparative experiments using **transition windows of different sizes**, and the results are shown below. We will include this experiment in the final version of the paper and revise our Motion In-between method accordingly. We sincerely appreciate your feedback, which has helped improve our approach.
>
>
> |Size| SmoothRate(Seen) $\downarrow$| SmoothRate(Unseen) $\downarrow$|
> |:-------:|:-------:|:-------:|
> | 1| 4.77 ± 0.68    |5.26 ± 2.47    | |
> | 3 | 3.59 ± 0.81   |4.58±1.05 |
> | 5 |**2.98 ± 0.43**   | **3.70 ± 0.61**|
> | 10 |3.24 ± 0.65    |4.08±0.82 |
> | 20 |3.30 ± 0.57    |4.19±0.74|

---

> > ### Comment · Reviewer_7WnN · 2025-08-06
> >
> > I appreciate the authors’ response. As most of my concerns have been addressed, I will maintain my positive score.

---

> > > ### Author Response · Authors · 2025-08-06
> > >
> > > We sincerely appreciate your valuable feedback and maintain your positive score. We are pleased to address your concerns and will further improve both the physical evaluation metrics and the Motion In-between Refinement module in the final version of the paper.

---

### Official Review · Reviewer_7v2M · 2025-07-02

**Clarity:** 3
**Significance:** 3
**Originality:** 3
**Rating:** 5
**Confidence:** 4

**Summary:**

OpenHOI is the first open-world hand-object interaction (HOI) synthesis framework that generates long-horizon manipulation sequences for unseen objects guided by open-vocabulary instructions. It fine-tunes a 3D multimodal large language model (MLLM) to support two key abilities: (1) Affordance prediction, localizing actionable regions on novel objects; and (2) Instruction decomposition, breaking complex commands into executable sub-tasks. The framework combines affordance reasoning with diffusion-based interaction generation and physics-aware refinement, achieving state-of-the-art performance and strong generalization across object categories, task structures, and language complexity.

**Questions:**

1. In line 193, shouldn't the term "hand joints" actually refer to hand surface vertices? If so, how did you recover the hand's surface?
2. In the affordance refinement, why is the distance computed to the nearest point on the entire object surface, rather than to the nearest point within the predicted contact region?

**Ethical Concerns:**

["NO or VERY MINOR ethics concerns only"]

**Final Justification:**

The paper makes a meaningful contribution to addressing HandOI, with well-motivated objectives and comprehensive experiments. The rebuttal has also resolved my concerns, so I am inclined to recommend acceptance.

**Limitations:**

yes

**Quality:**

3

**Strengths And Weaknesses:**

### **Strength**
- The paper is well-structured and easy to follow overall.
- Open-world HOI generation is a highly valuable and challenging direction. The paper effectively leverages the strengths of 3D MLLMs to provide affordance priors and integrates them with HOI generation, generating interactions under high-level instruction.
- Both the qualitative and quantitative results are promising, and the accuracy of the affordance prediction is also reasonably good.



### **Weakness**
- The visualizations provided for affordance prediction and hand-object interaction are somewhat limited, particularly for unseen objects. More qualitative examples would help better illustrate the model’s generalization capability.

---

> ### Author Rebuttal · Authors · 2025-07-31
>
> We greatly appreciate Reviewer 7v2M's valuable comments and insightful questions. We will address your questions in detail below.
>
> >Q1:The visualizations provided for affordance prediction and hand-object interaction are somewhat limited, particularly for unseen objects. More qualitative examples would help better illustrate the model’s generalization capability.
>
> A1: Thanks for your suggestion. We included additional visualizations of HOI sequences for unseen objects in the **Appendix(Figure A2)**. More visualizations of the affordance results will be added in the final version, particularly highlighting comparisons on unseen objects.
>
>
> >Q2:In line 193, shouldn't the term "hand joints" actually refer to hand surface vertices? If so, how did you recover the hand's surface?
>
> A2: Yes, you are right. The "hand joints" actually mean "hand surface vertices". We recover the hand's surface information to the MANO model using **$L_{hoi\_train}$ (Eq. (6))**.
>
>
> >Q3: In the affordance refinement, why is the distance computed to the nearest point on the entire object surface, rather than to the nearest point within the predicted contact region?
>
> A3: Thank you for raising the concern. We are sorry for the typo error in our description and will revise the explanation accordingly in the final version of the paper.  Yes, aligning the hand keypoints more closely with the affordance region yields superior performance. In our method, we explicitly guide the hand joints toward the **affordance region**, which is why we refer to this process as **"Affordance Refinement"**.

---

> > ### Comment · Reviewer_7v2M · 2025-08-04
> > **Comment by Reviewer 7v2M**
> >
> > Thank you for the clarification. I hope the authors will address the typos and provide improved visualizations in the final version. With that, my concerns have been resolved, and I will raise my rating for a clear acceptance of the paper.

---

> > > ### Author Response · Authors · 2025-08-05
> > >
> > > Thanks for your valuable suggestions and raising the rating. We will correct the typos and include additional improved visualizations in the final version.

---

### Note · Authors · 2025-08-15

We sincerely thank the reviewers for their careful reading and meaningful suggestions. We are encouraged that **all reviewers now express positive views of  OpenHOI**, with either increased ratings or maintained positive scores. They find our paper "highly valuable and challenging direction" (7v2M), "good motivation and strong method novelty" (7WnN), "significant contribution" (wJWx), "well written" (all reviewers) and most of their concerns are addressed.

Our rebuttal addressed the main concerns in the following key points:

- **Affordance as a key to Open-world HOI:** Open-world affordance is a **powerful and explicit prior** for interaction and serves as an important condition for the Affordance-driven diffusion model during both training and inference time, yielding strong **template-free** generalization capabilities in our model.
- **3D MLLM's Powerful Performance:**  ShapeLLM is a powerful 3D foundation model with strong performance on downstream tasks and is suitable for HOI synthesis. We use a **coarse-to-fine tuning** strategy: we first pre-train on object-centric affordance data to learn affordance priors, and then we fine-tune on HOI datasets to align semantics with the target domain, yielding open-world affordance grounding.
- **More experiments and results on OpenHOI:** Our rebuttal provides additional details on experiments and results, such as additional qualitative visualizations, physical evaluation, comparative experiments, more ablation studies, statistical significance tests, failure cases and error analyses.

OpenHOI is the first Open-world Hand-object Interaction Synthesis Framework. Positive reviewer feedback, rigorous methodology, and extensive experiments jointly indicate that OpenHOI constitutes a significant contribution to open-world hand–object interaction synthesis for long-horizon manipulation sequences with open-vocabulary instructions on unseen objects. We believe it will have broad applications in AR/VR, robotic manipulation and embodied AI in the real world.

---

### Decision · Program_Chairs · 2025-09-17

**Decision:**

Accept (oral)

**Comment:**

This paper introduces a novel framework for open-world 3D hand-object interaction (HOI) synthesis. Its core contribution is the integration of a fine-tuned 3D MLLM for affordance grounding and semantic task decomposition with an affordance-driven diffusion model for motion generation. This enables the system to generate long-horizon, physically plausible manipulation sequences for novel, unseen objects, addressing a significant limitation of prior closed-set methods.

This paper moves the field beyond closed-set constraints by creatively integrating the latest advances in 3D MLLMs with generative models. The proposed framework is novel, technically sophisticated, and delivers compelling empirical results that validate its core claims. This paper has the potential to significantly influence research in AR/VR, robotics, and embodied AI.